# A comprehensive synthetic library of poly-*N*-acetyl glucosamines enabled vaccine against lethal challenges of *Staphylococcus aureus*

Zibin Tan[1,2,3,10], Weizhun Yang [1,2,4,10], Nicholas A. O'Brien [5], Xingling Pan[1,2], Sherif Ramadan[1,2,6], Terence Marsh[7], Neal Hammer [7], Colette Cywes-Bentley[8], Mariana Vinacur[8], Gerald B. Pier [8], Jeffrey C. Gildersleeve[5] & Xuefei Huang[1,2,9] ✉

Poly-β-(1−6)-*N*-acetylglucosamine (PNAG) is an important vaccine target, expressed on many pathogens. A critical hurdle in developing PNAG based vaccine is that the impacts of the number and the position of free amine vs *N*-acetylation on its antigenicity are not well understood. In this work, a divergent strategy is developed to synthesize a comprehensive library of 32 PNAG pentasaccharides. This library enables the identification of PNAG sequences with specific patterns of free amines as epitopes for vaccines against *Staphylococcus aureus* (*S. aureus*), an important human pathogen. Active vaccination with the conjugate of discovered PNAG epitope with mutant bacteriophage Qβ as a vaccine carrier as well as passive vaccination with diluted rabbit antisera provides mice with near complete protection against infections by *S. aureus* including methicillin-resistant *S. aureus* (MRSA). Thus, the comprehensive PNAG pentasaccharide library is an exciting tool to empower the design of next generation vaccines.

Bacterial infections continue to threaten global health, and this situation is further exacerbated by the prevalence of antimicrobial-resistant strains including those resistant to multiple drugs. In 2019, the Center for Disease Control and Prevention (CDC) estimated that 3 million antimicrobial-resistant infections occurred in the USA annually[1]. Antibiotic resistance is rising to dangerously high levels in all parts of the world with resistance in some pathogens observed to nearly all antibiotics that have been developed[2]. New strategies to prevent and treat infections are urgently needed.

Parallel to the development of new antibiotics, vaccination is an important approach for combating pathogens[3]. Multiple anti-microbial

vaccines have been implemented against infections such as those by *Clostridium tetani*, *Bordetella pertussis*, and *Streptococcus pneumoniae*. However, despite these successes, there are no approved vaccines against many other deadly pathogens including *Staphylococcus aureus* (*S. aureus*), which caused over 300,000 infections and 20,000 deaths through bloodstream infection in 2017 in the USA[4]. The prevalence of methicillin-resistant *S. aureus* (MRSA) further highlights the need for a vaccine to stem the rise of anti-microbial resistance[5].

A major challenge in vaccine design is the selection of a suitable antigen. The cell wall of *S. aureus* as well as numerous other bacterial, fungal, and protozoal parasites contains the polysaccharide

[1]Department of Chemistry, Michigan State University, 578 S. Shaw Lane, East Lansing, MI 48824, USA. [2]Institute for Quantitative Health Science and Engineering, Michigan State University, East Lansing, MI 48824, USA. [3]Center for Cancer Immunology, Faculty of Pharmaceutical Sciences, Shenzhen Institute of Advanced Technology, Chinese Academy of Sciences (CAS), Shenzhen, Guangdong 518000, China. [4]School of Chemistry and Materials Science, Hangzhou Institute for Advanced Study, University of Chinese Academy of Sciences, Hangzhou, Zhejiang 310024, China. [5]Chemical Biology Laboratory, Center for Cancer Research, National Cancer Institute, Frederick, MD 21702, USA. [6]Chemistry Department, Faculty of Science, Benha University, Benha, Qaliobiya 13518, Egypt. [7]Department of Microbiology, Genetics & Immunology, Michigan State University, East Lansing, MI 48824, USA. [8]Division of Infectious Diseases, Department of Medicine, Brigham and Women's Hospital, Harvard Medical School, Boston, MA 02115, USA. [9]Department of Biomedical Engineering, Michigan State University, East Lansing, MI 48824, USA. [10]These authors contributed equally: Zibin Tan, Weizhun Yang. ✉e-mail: huangxu2@msu.edu

poly-β-(1–6)-*N*-acetylglucosamine (PNAG), which is composed of β-(1–6) linked glucosamine units with 80–95% of them *N*-acetylated[6–9]. Naturally occurring PNAG can vary in its chemical structures with the amino groups in some glucosamine existing as free amines rather than as *N*-acetamides (NHAc)[10]. The differing degrees and positions of free amines vs NHAc in PNAG can result in high structural heterogeneity, and the precise PNAG structures from bacteria are not known. Present on the cell surface and as an integral component of the biofilm, PNAG has been found to be an important virulence factor that aids the bacterial evasion of the immune system[6,11]. The widespread expression of PNAG in multiple pathogenic microbes and its important roles in pathogenesis render it an attractive target for vaccine development.

The immunological properties of PNAG have been investigated[6,12]. PNAG pentasaccharides and nonasaccharides with all amine groups either free or fully acetylated have been synthesized and subsequently conjugated with an immunogenic protein carrier, tetanus toxoid (TT)[13–15]. Mouse immunization studies showed that the TT conjugates with the PNAG bearing all amines could induce protective immunity, while the antibodies elicited by the TT conjugates with fully acetylated PNAG counterparts failed to mediate protective functions such as opsonic killing and in vivo protection[12]. Despite this interesting finding, PNAG-based vaccines investigated to date only contained the glycans with all amine groups either free or fully acetylated. It is not clear what PNAG structure would comprise the best epitope(s) for maximal protective immunity, and whether specific patterns of amine vs acetylation in PNAG can be designed to enhance vaccine efficacy.

To gain a deeper understanding of the impact of variably acetylated PNAG epitopes, diverse structurally defined PNAG sequences are critically needed. In this work, we report a divergent strategy enabling the synthesis of a comprehensive library of 32 PNAG pentasaccharides with all possible combinations of the location and the number of free amines incorporated into the oligosaccharide. The availability of such a library enables us to establish the amine/acetylation code (locations of the free amine/NHAc in PNAG) of an anti-PNAG monoclonal antibody (mAb). The fine patterns of free amine/NHAc of PNAG oligosaccharides are found to be critical for mAb binding. The structural patterns identified through the microarray study have guided the selection of PNAG epitopes for the design for next generation vaccine, which provides highly effective protection in multiple mouse models against *S. aureus* infections, including those by MRSA.

## Results

### Synthesis of the PNAG oligosaccharide library

To date, only fully acetylated and fully deacetylated PNAG oligosaccharides have been investigated as immunogens for vaccine studies[13–15]. The availability of a library of PNAG oligosaccharides with systematically varied numbers and locations of free amines can greatly aid in the identification of the maximally protective epitope structures. We aimed to synthesize a comprehensive library of 32 pentasaccharides designated PNAG0–PNAG31 (Fig. 1) fully covering the free amine space of PNAG. The reducing ends of the target pentasaccharides bear a linker terminated with a disulfide group, which can be reduced for chemoselective conjugation to a carrier protein.

While several PNAG structures have been synthesized before[16–20], a general method for the expeditious construction of a comprehensive PNAG pentasaccharide library is lacking. To accelerate the library synthesis, rather than starting from monosaccharide building blocks for each targeted pentasaccharide, we envisioned the overall efficiency can be significantly enhanced with a divergent strategy. In our synthetic design, the amine groups of strategically protected pentasaccharides are differentiated by orthogonal protective groups for selective deprotection and acetylation. After screening multiple synthetic intermediates, we developed two key linchpin pentasaccharide intermediates (**1** and **2**), which bear four protective groups, i.e., *tert*-butyloxycarbonyl (Boc), allyloxycarbonyl (Alloc), 2,2,2-trichloroethoxycarbonyl (Troc), and fluorenylmethoxycarbonyl (Fmoc), on glucosamine units A, B, C, and D. The reducing end glucosamine unit E is *N*-acetylated (for compound **1**) or *N*-trifluoroacetylated (for compound **2**) (Fig. 2A).

Based on the above design, our synthesis commenced from thioglycoside **3**, which glycosylated 3-azido-1-propanol **4** to provide compound **5** in 82% yield (Fig. 2B). Upon removal of the Alloc group from **5** and *N*-acetylation, the resulting compound **6** was subjected to azide reduction, amidation of the free amine with carboxylic acid **7**,

**Fig. 1 | Structures of the comprehensive library of PNAG pentasaccharides.** The five-digit number in the bracket for each compound codes for free amine (0) or *N*-acetamide (1) at residues ABCDE from the non-reducing end to the reducing end of the pentasaccharide, respectively. The five-digit number was then viewed as a binary number and converted to the decimal system as the compound number. For example, 01010 in binary number is equivalent to 10 in the decimal system. Thus, the PNAG pentasaccharide bearing *N*-acetylation at units B and D only is named as PNAG10.

PNAG0 (00000): R₁, R₂, R₃, R₄, R₅ = H;
PNAG1 (00001): R₁, R₂, R₃, R₄ = H; R₅ = Ac
PNAG2 (00010): R₁, R₂, R₃, R₅ = H; R₄ = Ac
PNAG3 (00011): R₁, R₂, R₃ = H; R₄, R₅= Ac
PNAG4 (00100): R₁, R₂, R₄, R₅ = H; R₃ = Ac
PNAG5 (00101): R₁, R₂, R₄ = H; R₃, R₅ = Ac
PNAG6 (00110): R₁, R₂, R₅ = H; R₃, R₄ = Ac
PNAG7 (00111): R₁, R₂ = H; R₃, R₄, R₅ = Ac
PNAG8 (01000): R₁, R₃, R₄, R₅ = H; R₂ = Ac
PNAG9 (01001): R₁, R₃, R₄ = H; R₂, R₅ = Ac
PNAG10 (01010): R₁, R₃, R₄ = H; R₂, R₄ = Ac
PNAG11 (01011): R₁, R₃ = H; R₂, R₄, R₅= Ac
PNAG12 (01100): R₁, R₄, R₅ = H; R₂, R₃ = Ac
PNAG13 (01101): R₁, R₄ = H; R₂, R₃, R₅ = Ac
PNAG14 (01110): R₁, R₅ = H; R₂, R₃, R₄ = Ac
PNAG15 (01111): R₁ = H; R₂, R₃, R₄, R₅ = Ac

PNAG16 (10000): R₂, R₃, R₄, R₅ = H; R₁ = Ac
PNAG17 (10001): R₂, R₃, R₄ = H; R₁, R₅ = Ac
PNAG18 (10010): R₂, R₃, R₅ = H; R₁, R₄ = Ac
PNAG19 (10011): R₂, R₃ = H; R₁, R₄, R₅ = Ac
PNAG20 (10100): R₂, R₄, R₅ = H; R₁, R₃ = Ac
PNAG21 (10101): R₂, R₄ = H; R₁, R₃, R₅ = Ac
PNAG22 (10110): R₂, R₅ = H; R₁, R₃, R₄ = Ac
PNAG23 (10111): R₂ = H; R₁, R₃, R₄, R₅ = Ac
PNAG24 (11000): R₃, R₄, R₅ = H; R₁, R₂ = Ac
PNAG25 (11001): R₃, R₄ = H; R₁, R₂, R₅ = Ac
PNAG26 (11010): R₃, R₄ = H; R₁, R₂, R₄ = Ac
PNAG27 (11011): R₃ = H; R₁, R₂, R₄, R₅= Ac
PNAG28 (11100): R₄, R₅ = H; R₁, R₂, R₃ = Ac
PNAG29 (11101): R₄ = H; R₂, R₁, R₃, R₅ = Ac
PNAG30 (11110): R₅ = H; R₂, R₁, R₃, R₄ = Ac
PNAG31 (11111): R₁, R₂, R₃, R₄, R₅ = Ac

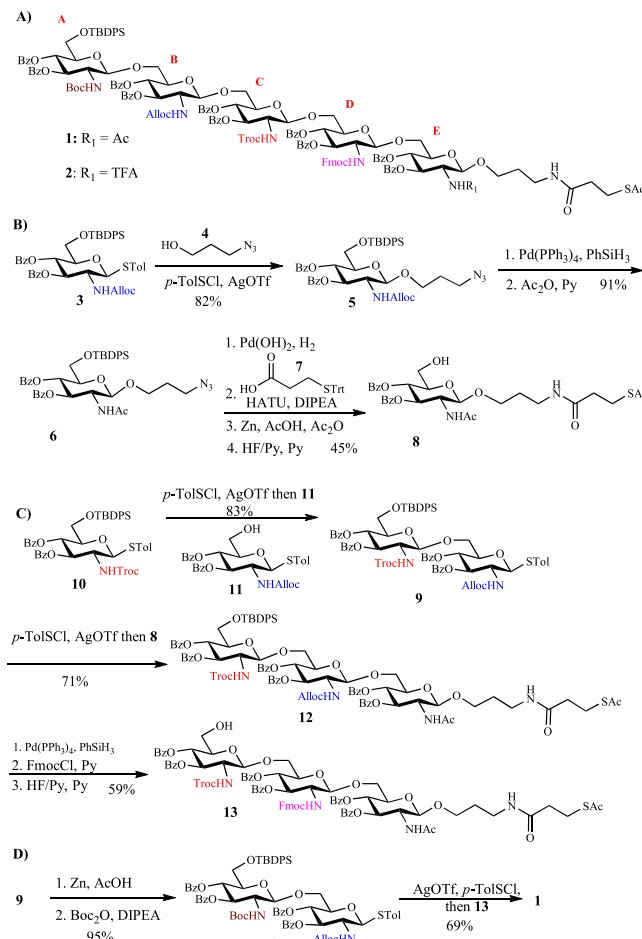

**Fig. 2 | Structures and syntheses of key intermediates. A** Structures of two key linchpin pentasaccharide intermediates (**1** and **2**). **B** Synthesis of the reducing end glucosamine building block **8**. **C** Syntheses of compound **13**; **D** Syntheses of compound **1**. Ac acetyl, Alloc allyloxycarbonyl, Bz benzoyl, TBDPS *tert*-butyldiphenylsilyl, Boc *tert*-butyloxycarbonyl, DIPEA diisopropylethylamine, Fmoc fluorenylmethoxycarbonyl, HATU hexafluorophosphate azabenzotriazole tetramethyl uronium, and Troc 2,2,2-trichloroethoxycarbonyl.

and protective group adjustments leading to compound **8** in 45% yield for the four steps.

Oligosaccharide assembly started from the CD disaccharide **9** containing *N*-Troc and *N*-Alloc groups (Fig. 2C). Thioglycoside donor **10** was preactivated with the *p*-TolSCl/AgOTf promoter system[21] at −78 °C. Upon complete activation, the thioglycosyl acceptor **11** was added to the reaction mixture leading to disaccharide **9** in 83% yield (Fig. 2C). In order to extend the glycan chain, the glycosylation of acceptor **8** with disaccharide **9** was performed. When the reaction was first carried out under the pre-mix condition, i.e., **9** and **8** were mixed together followed by the addition of promoter (*p*-TolSCl/AgOTf or NIS/TfOH[22,23]), little desired trisaccharide **12** was obtained, which was likely due to the activation of the thioester moiety by the thiophilic promoter. Next, the reaction was explored under the pre-activation condition by activating **9** with the promoter *p*-TolSCl/AgOTf first, followed by the subsequent addition of acceptor **8**. This change of the reaction protocol successfully produced trisaccharide **12** in 71% yield. Replacement of Alloc with Fmoc and removal of TBDPS group from **12** resulted in the trisaccharide **13**. To extend **13** to a pentasaccharide, the Troc moiety of disaccharide **9** was replaced with Boc (disaccharide **14**, Fig. 2D). Pre-activation-based glycosylation of **14** and **13** produced pentasaccharide **1**, which contains four different *N*-protective groups on units A, B, C, and D. Analogously, pentasaccharide **2** was

synthesized with four different *N*-protective groups on units A, B, C, and D, and the *N*-TFA group on unit E (Supplementary Fig. 1).

With the two key pentasaccharides in hand, we explored orthogonal deprotection of pentasaccharides **1** and **2**. As an example, the Boc and Alloc groups of compound **2** could be removed by 90% aqueous TFA and Pd(PPh₃)₄/PhSiH₃, while deprotections of Troc and Fmoc were accomplished using Zn/AcOH and 20% piperidine in *N,N*-dimethylformamide (DMF), respectively, without affecting any other amine protective groups (Fig. 3A). These results suggest that the four amine protective groups could be independently removed specifically.

With the orthogonal deprotection conditions established, divergent modifications of the key pentasaccharide intermediates were carried out. Treatment of pentasaccharide **1** with 90% TFA cleaved both Boc and TBDPS groups (Fig. 3B). Upon acetylation of the newly liberated hydroxyl and amine moieties, the Alloc, Troc, and Fmoc groups were subsequently removed followed by full *O*- and *S*-deacylation with 20% hydrazine hydrate in MeOH, affording PNAG17 pentasaccharide in 48% overall yield bearing the *N*-acetylglucosamine (GlcNAc)-glucosamine (GlcN)-GlcN-GlcN-GlcNAc (10001) sequence. Alternatively, following TFA treatment of **1**, the Fmoc group was cleaved, which was then acetylated with subsequent removal of Troc, Alloc, and Bz moieties to produce pentasaccharide PNAG19 with the GlcNAc-GlcN-GlcN-GlcNAc-GlcNAc sequence (10011) in 51% overall yield. Similar divergent modification processes on the two key pentasaccharides **1** and **2** produced the full library of 32 PNAG pentasaccharides with all possible combinations of free amines in each glucosamine unit of the pentasaccharides (Fig. 3B, C).

**Superiority of the mQβ−PNAG in inducing anti-PNAG antibodies**

As carbohydrate antigens in general are T cell independent B cell antigens[24] and small oligosaccharides alone are not immunogenic[25], these types of antigens need to be conjugated to an immunogenic carrier in order to induce anti-carbohydrate IgG antibody responses. The mutant bacteriophage Qβ (mQβ)[26] is a powerful carrier and likely highly useful for carbohydrate based conjugate vaccines[27–29]. As PNAG oligosaccharides can potentially contain multiple free amine moieties, we resorted to sulfhydryl chemistry for PNAG/mQβ conjugation. The mQβ A38K/A40C/D102C was expressed in *E. coli*, purified, and incubated with the bifunctional linker succinimidyl 3-(bromoacetamido) propionate (SBAP) **19** to react with free amines on the mQβ surface (Fig. 4A). Upon removal of the excess linker, the SBAP functionalized mQβ was added to the PNAG pentasaccharide followed by quenching the unreacted bromoacetamide moieties on mQβ with cysteine to avoid any potential side reactions of residual bromoacetamide on mQβ upon storage or following vaccination. MALDI-TOF mass spectrometry (MS) analysis of the mQβ−PNAG conjugate showed an average loading of 250 copies of pentasaccharide per particle (Supplementary Fig. 2A). It is known that the antigen loading density on Qβ can significantly impact the levels of antibodies induced against the target antigen[29,30]. When the loading level of antigen was low (<50 copies per particle), despite the same total amount of antigen administered, the antibody responses induced were low[29,30]. Increasing the local density of the antigen on the particle (over 100 antigens per particle) can significantly improve the antibody responses, which is presumably due to the more effective crosslinking of the B cell receptors on B cells[31]. The loading density of PNAG on the mQβ−PNAG conjugate was higher than the threshold antigen level needed for powerful B cell activation.

With the mQβ−PNAG conjugates in hand, their abilities to induce anti-PNAG antibodies were evaluated. The conjugate of TT with the PNAG pentasaccharide bearing five free amines (5GlcNH₂−TT) has undergone a phase 1 human clinical trial[32]. To compare with our mQβ−PNAG conjugate, we covalently linked PNAG0 (5GlcNH₂) with the TT heavy chain (TTHc) using SBAP and achieved an average loading of 4.7 PNAG0 per protein molecule (Fig. 4B and Supplementary

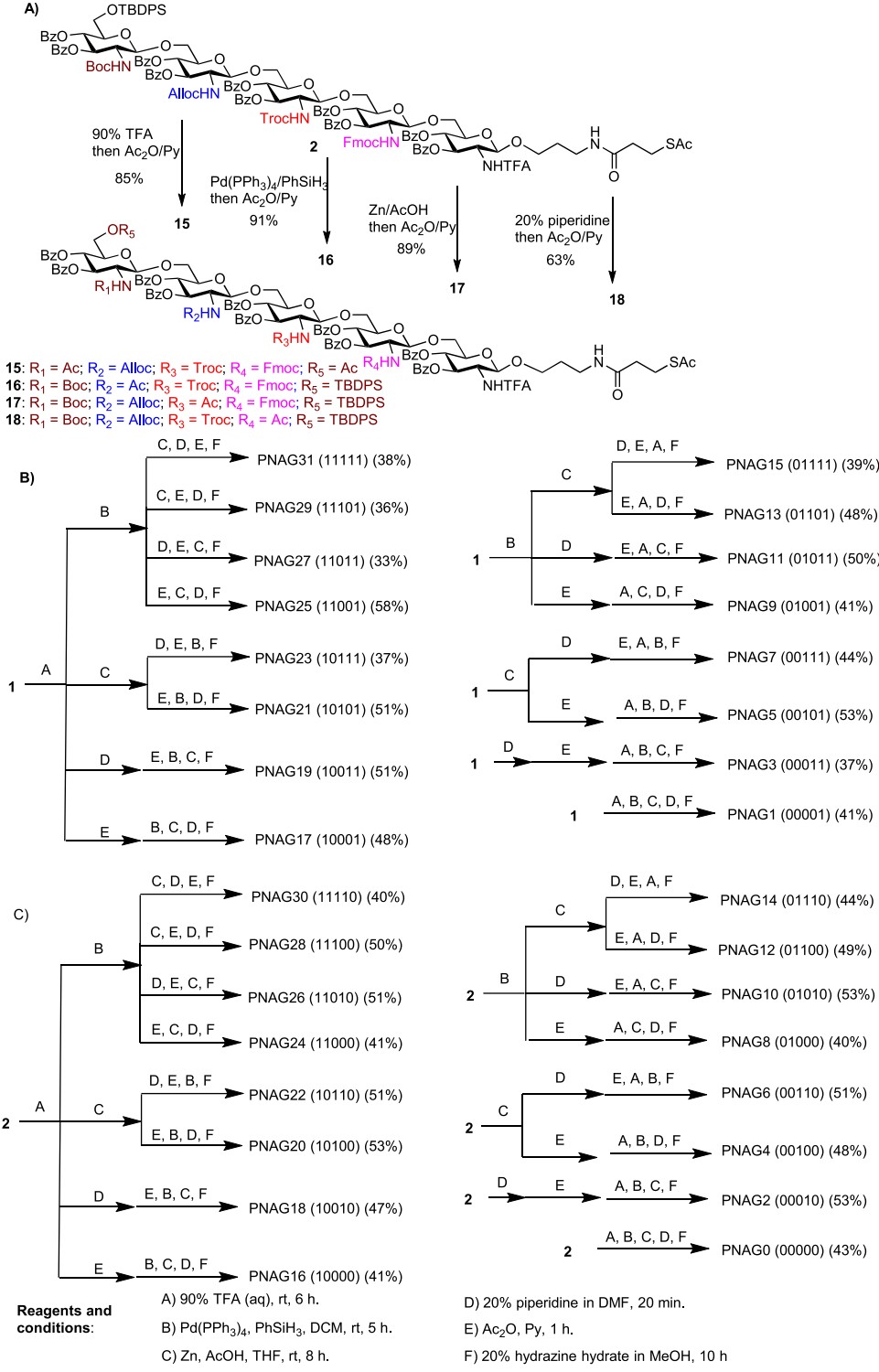

**Fig. 3 | Synthesis of the 32 membered comprehensive PNAG pentasaccharide library. A** Orthogonal deprotection of pentasaccharide **2**. **B** Divergent syntheses of 16 PNAG pentasaccharides from the strategically protected pentasaccharide **1**. **C** Divergent syntheses of 16 PNAG pentasaccharides from the strategically protected pentasaccharide **2**. Ac acetyl, Alloc allyloxycarbonyl, Bz benzoyl, TBDPS *tert*-butyldiphenylsilyl, Boc *tert*-butyloxycarbonyl, DIPEA diisopropylethylamine, DMF dimethylformamide, Fmoc fluorenylmethoxycarbonyl, HATU hexafluorophosphate azabenzotriazole tetramethyl uronium, Troc 2,2,2-trichloroethoxycarbonyl, and TFA trifluoroacetic acid.

Fig. 2B). The recombinant TTHc is a suitable surrogate of TT[33]. As the molecular weight of mQβ particle (2540 kDa for the protein shell) is about 49 times that of the TTHc (MW-52 kDa), the overall densities of PNAG0 on mQβ–PNAG0 and TTHc–PNAG0 were similar.

Head-to-head comparative immunogenicity studies of the mQβ–PNAG0 and the TTHc–PNAG0 conjugates were carried out.

Groups of female C57Bl6 mice ($n = 5$ per group) were immunized with freshly prepared mQβ–PNAG0 (8 nmol corresponding to 8 μg of PNAG0 per injection) or the TTHc–PNAG0 conjugate (8 nmol PNAG0 per injection) on days 0, 14, and 28. Monophosphoryl lipid A (MPLA, 20 μg) was added to each vaccination as the adjuvant. A control group of mice received a mixture of mQβ with PNAG0 at equivalent total

amounts of mQβ, PNAG0, and MPLA following the same immunization protocol. On day 35, sera were collected from all mice.

To analyze the levels of antibodies generated, enzyme linked immunosorbent assay (ELISA) analyses were performed. To avoid the interference of anti-mQβ antibodies in the sera, the 32 PNAG pentasaccharides were conjugated with BSA individually (Fig. 4C and Supplementary Fig. 3) and used as the ELISA coating antigens. As shown in Fig. 5A, mQβ–PNAG0 induced high anti-PNAG IgG titers (EC50 IgG titers GMT 75,613, measured against BSA–PNAG0) while the IgM titers were negligible (GMT < 1000). Furthermore, high levels of anti-PNAG0 IgG responses were observed more than 1 year after the initial immunization (Fig. 5B). The IgG levels could be boosted back to near peak levels after nearly 2 years indicating that the mQβ conjugate induced PNAG0 specific memory B cells through immunization. The GMT of 75,613 achieved in mice receiving the mQβ–PNAG0 was significantly (P < 0.0001, Dunnett's multiple comparisons test) higher than the anti-PNAG0 IgG titers achieved in mice immunized with the corresponding TTHc–PNAG0 conjugate (GMT 4765), *highlighting the superior immunogenicity of the mQβ carrier for conjugate vaccines*. Mice immunized

with the admixture of mQβ and PNAG0 did not produce any detectable levels of anti-PNAG0 IgG (GMT < 1000), accentuating the critical need to covalently conjugate mQβ with PNAG0.

As C57Bl6 mice are inbred, to enhance the rigor of our study, we immunized outbred CD1 mice with the mQβ–PNAG0 conjugate following the same immunization protocol. mQβ–PNAG0 was able to elicit comparably high titers of anti-PNAG0 IgG antibodies on day 35 after the primary series of immunization in CD1 mice (Supplementary Fig. 4).

## Probing the epitope specificity of an anti-PNAG mAb

The precise PNAG sequences synthesized by pathogens such as *S. aureus* are not known. Furthermore, the most abundant PNAG structure on cell surfaces that would encompass a highly (80–95%) acetylated polysaccharide is not a protective epitope[13]. To guide vaccine design, we envisioned anti-PNAG mAb F598 could provide valuable information regarding optimal acetylation patterns in a PNAG pentasaccharide. Isolated from a patient who recovered from an *S. aureus* infection, mAb F598 can protect mice against *S. aureus* infections[34]. The 32 PNAG pentasaccharide–BSA conjugates were immobilized onto a glycan microarray[35]. Following incubation of mAb F598 with the microarray and washing, the amount of antibody remaining bound was quantified with a fluorescent secondary antibody. Interestingly, although mAb F598 was initially identified due to binding to deacetylated PNAG with only ~15% N-acetylation[34], it had little binding to glycan PNAG0 or any glycans containing only one Ac moiety. Highly acetylated PNAG such as PNAG30 and PNAG31 with four or more consecutive GlcNAcs were among the strongest binders (Fig. 6). Both the location and the number of NHAc are important for F598 binding, supporting the idea of an amine/acetylation code. For example, despite having the same total number of NHAcs (4 in the molecules), PNAG23 (10111) is a weak binder with an apparent affinity <5% of that with PNAG30 (11110). Out of the PNAGs with two or three GlcNAc residues, PNAG10 and PNAG26 were the strongest binders, respectively.

To better interpret the binding data, we quantified the GlcNAc binding preference of F598 by computing the preference index $P$ for each unit of the pentasaccharide as

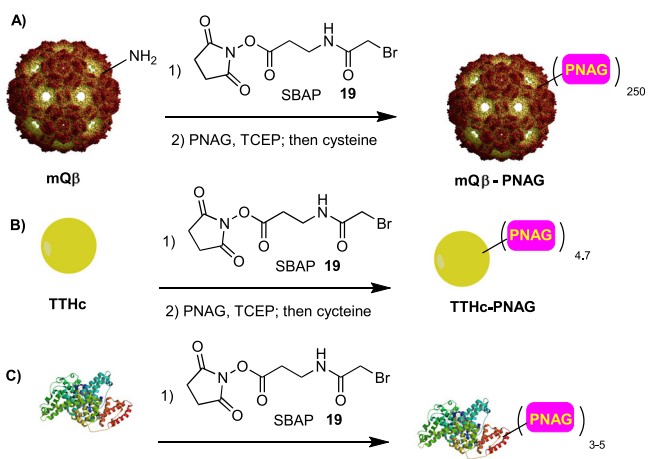

**Fig. 4 | Syntheses of conjugates.** Syntheses of **A** mQβ–PNAG, **B** TTHc–PNAG, and **C** BSA–PNAG conjugates. SBAP succinimidyl 3-(bromoacetamido)propionate, TCEP tris(2-carboxyethyl)phosphine, TTHc tetanus toxoid heavy chain.

$$P_i = \frac{\sum_j R_j \times A_i}{\sum_j R_j} \qquad (1)$$

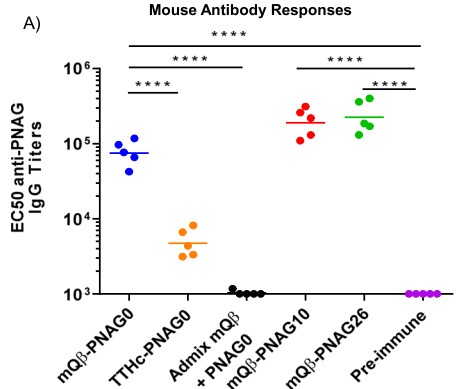

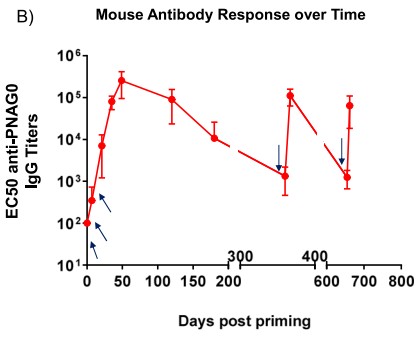

**Fig. 5 | Immunization of mice with mQβ–PNAG led to high levels and long lasting anti-PNAG IgG antibodies. A** C57Bl6 mouse (*n* = 5 per group) antibody responses at day 35 after immunization. The EC50 value (the fold of serum dilution that gives half-maximal binding) of the IgG titers to the immunizing oligosaccharide was plotted with each symbol representing one animal and the horizontal line is the geometric mean value of the titers within the group. The ELISA titers were determined using the BSA–PNAG conjugate containing the same PNAG structure as the immunizing Qβ–PNAG construct. One-way ANOVA allowed for rejection of the null

hypothesis that all groups have the same mean IgG titers (*P* < 0.0001). Statistical significance was performed by Dunnett's multiple comparisons post-hoc test. ****P < 0.0001; **B** Anti-PNAG0 IgG antibody responses of mice (*n* = 5) immunized with mQβ–PNAG0 monitored over time with mean titers plotted. Data are presented as mean values ± standard deviation of the titer numbers from five mice. The arrows indicate days of vaccination (days 0, 14, 28, 360, and 655). The antibody responses could be boosted more than 650 days after prime vaccination. Source data are provided as a Source Data file.

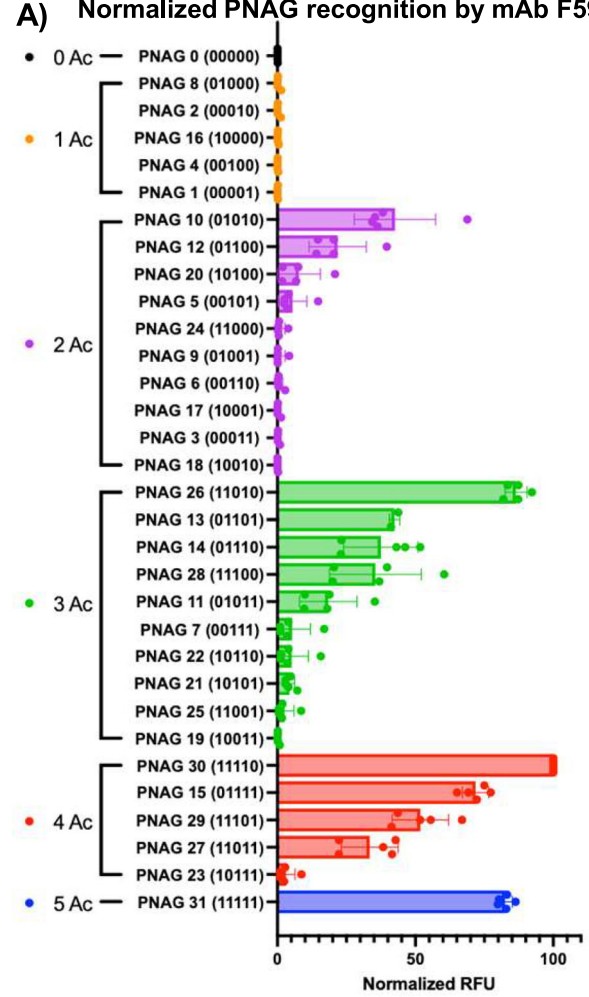

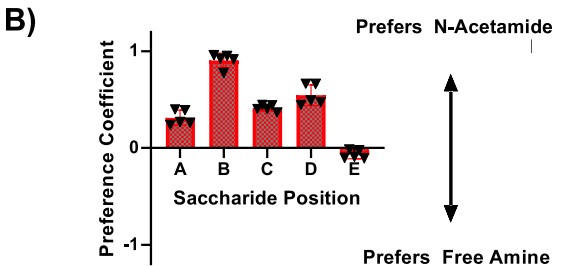

**Fig. 6 | Determination of the epitope profile of F598 mAb with the glycan microarray. A** Relative fluorescence unit (RFU) of F598 mAb binding with the library of 32 PNAG pentasaccharides. The glycans are grouped according to the number of NHAc units in the molecule. Each glycan sequence is printed five times on the glycan microarray. The error bars represent the standard deviations of five individual spots. Data are presented as mean values ± standard deviation. F598 generally prefers highly acetylated PNAG sequences. Both the location and the number of NHAc units are important determinants of F598 binding.
**B** Quantification of the preference of F598 for acetylation at each site of the PNAG pentasaccharide. The mean values are calculated from the values of the binding intensities of all 32 PNAG sequences to F598. Each PNAG sequence is printed five times on the glycan microarray. Data are presented as mean values ± standard deviation. Source data are provided as a Source Data file.

where $i$ (A–E) is the site of monosaccharide from the non-reducing end to the reducing end, $j$ (0–31) is the serial number of glycan, $R$ is the intensity of the binding signal (RLU), and $A$ is the code for amine vs

acetylation ($A = -1$ for free amine and $A = 1$ for NHAc). $P$ value indicates the conditional probability difference between finding an NHAc or free amine for binding, which ranges from −1 to 1 with −1 and 1 indicating a complete preference for free amine or NHAc, respectively, at the specific site. As shown in Fig. 6B, unit B position showed the highest $P$ value of 0.91, suggesting on average that there is a 95.5% chance to find an NHAc moiety rather than a free amine on saccharide B for ligand binding with F598. The $P$ values for sites A, C, and E were between 0.31 and 0.54 indictive of a moderate global preference for $N$-acetylation. There were almost no preferences for NHAc or free amine for site 5 as the $P$ value at this site was close to 0.

The importance of an NHAc at unit B identified from microarray binding can be rationalized by the crystal structure of F598 complexed with fully acetylated PNAG oligosaccharides (PDB 6be4)[36]. The binding pocket of F598 could accommodate PNAG with five GlcNAc residues. The NHAc groups on saccharides B and D in the binding pocket were deeply inserted into the groove clamped by the heavy and the light chain of the mAb, forming multiple hydrogen bonds, while the NHAcs on units A, C, and E only had weak to moderate interactions with the antibody. The carbonyl oxygen of the NHAc on saccharide B forms a hydrogen bonding with light chain A32 backbone amide while bridging with light chain R52 residue via a water molecule. The carbonyl oxygen of NHAc on saccharide D also formed hydrogen bonds with light chain A97 backbone amide and the hydroxyl of heavy chain Y50. Those interactions supported the relatively high dependence of NHAc on sites B and D for the binding of F598.

**Critical PNAG sequence for high immunogenicity of mQβ–PNAG**
Based on the microarray results and the report that antibodies raised against the fully acetylated PNAG antigen were poorly protective[13–15], we selected PNAG10 and PNAG26 as new PNAG oligosaccharide antigens for further evaluations. PNAG10 has the strongest binding to F598 among all PNAG structures with two or fewer NHAcs, and PNAG26 is the best binder among all structures with three or fewer NHAcs. Both PNAG10 and PNAG26 have NHAcs on glycan sites B and D. PNAG0 was utilized as a positive control since the corresponding TT–PNAG0 construct (5GlcNH$_2$–TT) has entered clinical trials [ClinicalTrials.gov Identifier: NCT02853617].

C57/Bl6 mice were immunized with the mQβ conjugates of PNAG10 or PNAG26 following the aforementioned immunization protocol (8 nmol PNAG, three injections on days 0, 14, and 28 with MPLA adjuvant). ELISA analysis of the immune sera showed significantly enhanced IgG antibody titers against the immunizing antigen (PNAG10 or PNAG26) with GMT's of 191,141 and 227,064 ELISA units, respectively, as compared to pre-immune sera (Fig. 5A). Similarly, mQβ–PNAG10 or PNAG26 conjugates induced high levels of anti-PNAG10 and anti-PNAG26 IgG antibodies, respectively, in CD1 mice (Supplementary Fig. 4).

To demonstrate the immunogenicity of the mQβ conjugates in an additional mammalian species, New Zealand white rabbits were immunized with mQβ conjugates of PNAG0, PNAG10, and PNAG26 (8 nmol PNAG per injection) following a similar prime-boost protocol as that used in the mouse study. ELISA analysis of the post-immune sera showed that all three constructs induced strong anti-PNAG IgG responses with EC50 titers over 100,000 ELISA units (Fig. 7A), while those for the pre-immune sera were below 1000 ELISA units. No side effects due to vaccinations were observed in either rabbits or mice.

We analyzed next the recognition of native PNAG using PNAG polysaccharide isolated from *Acinetobacter baumannii*[37,38] as the coating antigen for ELISA. As shown in Fig. 7B, control sera from rabbits immunized only with the Qβ carrier did not bind with PNAG. In contrast, sera from rabbits immunized with mQβ–PNAG0, mQβ–PNAG10, and mQβ–PNAG26 exhibited strong binding with mQβ–PNAG26 antiserum having the highest titer (1,584,983 ELISA units) to the native microbial polysaccharide. As a comparison, sera

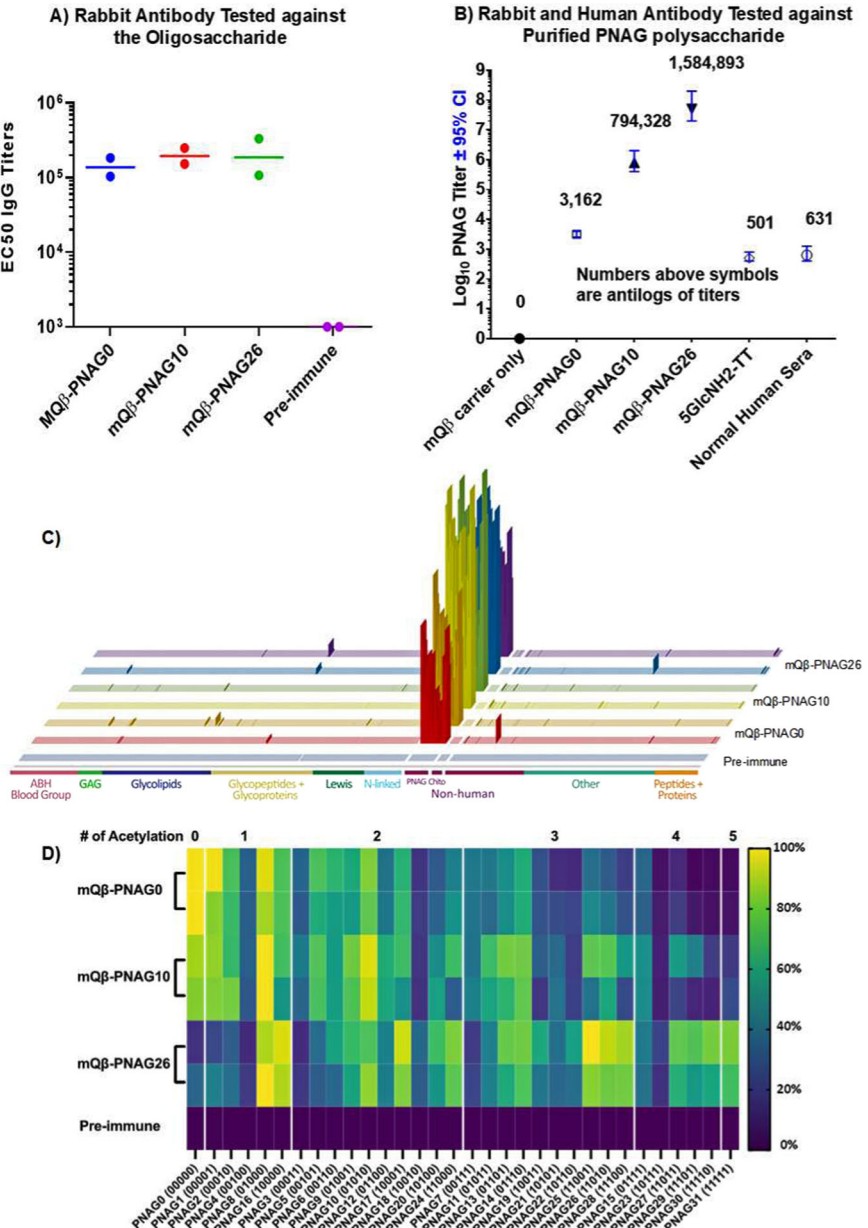

**Fig. 7 | Immunization of rabbits with mQβ–PNAG conjugate induced significant anti-PNAG IgG antibodies. A** IgG antibody titers to the immunizing PNAG oligosaccharide in rabbit (*n* = 2 per group) sera on day 35 after prime vaccination. **B** IgG antibody titers in pooled rabbit sera from mQβ-conjugate or 5GlcNH₂-TT conjugate immunized animals (*n* = 2 per group) as well as titer of natural human IgG in pooled human serum against native PNAG polysaccharide purified from *Acinetobacter baumannii*. The numbers above symbols are the average titer numbers. Titers and 95% confidence intervals (CI) were determined by linear regression using log₁₀ values of the average of replicate serum dilutions to determine the *X* intercept and 95% CI when *Y* = 0.5 (OD₄₀₅ nm of ELISA plate reading). **C** Stacked bar graphs depicting the IgG signals at the serum dilution of 1:50,000 for each rabbit (*n* = 2)

immunized with mQβ–PNAG0, mQβ–PNAG10, and mQβ–PNAG26 as well as pre-immune sera, respectively, on the array. The complete microarray results are provided in the Source Data file; **D** Normalized binding of the comprehensive library of PNAG pentasaccharides by IgG antibodies from post-immune sera of rabbits immunized with mQβ–PNAG0, mQβ–PNAG10, and mQβ–PNAG26, respectively, as well as pre-immune sera. PNAG sequences are grouped together according to the total number of acetamides in the molecules. The color scale bar is shown on the right with 100% indicating the strongest binding to a PNAG component and 0% indicating the weakest binder. For each antigen, the two rows represent sera from two rabbits per group immunized with the specific construct. Source data are provided as a Source Data file.

from the conjugate of 5GlcNH₂–TT[13,15] immunized rabbit only gave a titer of 501 ELISA units (Fig. 7B). Normal human sera containing natural antibodies to PNAG had an average ELISA titer of 631 ELISA units. These results further highlight the potential of mQβ–PNAG conjugates as vaccines.

Analysis of the microarray binding by post-immune sera revealed selective PNAG epitope recognition by the post-immune sera (Fig. 7D). Rabbits immunized with mQβ–PNAG0 produced serum IgG antibodies exhibiting the strongest binding with the immunizing PNAG0

antigenic structure. Other good binders include PNAG1 and PNAG8, both having a single GlcNAc in the structure. Interestingly, for PNAG4 with the sequence of GlcN-GlcN-GlcNAc-GlcN-GlcN, although it also only contains one GlcNAc, it had much lower binding with the sera (about 30% that to PNAG1). This suggests that three or more consecutive GlcNs are important for binding by anti-PNAG0 sera.

mQβ–PNAG10 immunized rabbits produced serum antibodies that preferentially bind to PNAG8 (01000) and PNAG10 (01010), which differ only by the GlcNAc in residue D indicating the non-reducing end

GlcN-GlcNAc-GlcN may be the main epitope. Serum antibodies from mQβ–PNAG26 (11010) immunized rabbits preferentially bound to PNAG25 (11001), PNAG26 (11010), PNAG8 (01000), and PNAG16 (10000) suggesting GlcNAc-GlcNAc-GlcN and GlcNAc-GlcN-GlcN may be part of the epitopes being recognized.

## Antisera-mediated bacterial recognition and killing

For an effective vaccine, it is important to establish that the post-immune sera bind not only the isolated antigen but also the antigen expressed on pathogen cells. We reacted *S. aureus* ATCC29213 cells with rabbit immune sera and the bound antibodies were detected by a fluorescently labeled anti-rabbit IgG secondary antibody. As shown in Supplementary Fig. 5A, fluorescence microscopy images showed stronger binding to bacterial cells by IgG antibodies in mQβ–PNAG10 and mQβ–PNAG26 immune sera compared to sera from mQβ–PNAG0 immunized rabbits or pre-immune sera. To validate pathogen recognition observed in fluorescence images, whole cell ELISA was performed. *S. aureus* cells were coated on ELISA plates, incubated with rabbit immune sera, and detected by secondary antibodies. The post-immune sera exhibited significantly higher titers in binding with the cells compared to pre-immune sera (Supplementary Fig. 6).

For antibody-mediated complement deposition[39], we added various immune sera to wells coated with purified PNAG isolated from *Acinetobacter baumannii*[37,38] along with IgG/IgM depleted 2.5% human complement (Fig. 8A). After incubation, the immobilized complement component C1q was detected by anti-C1q antibodies. As shown in Fig. 8A, sera from mQβ–PNAG10 and mQβ–PNAG26 deposited significantly more C1q than those from mQβ–PNAG0 immunized rabbits, which in turn had more potent C1q binding than antibodies in sera from rabbits immunized with the 5GlcNH₂–TT conjugate[13,15].

The abilities of the post-immune sera to kill bacteria in vitro were evaluated next via the opsonophagocytic killing (OPK) assay. *S. aureus* cells were treated with pooled rabbit immune sera, followed by the addition of complement/phagocytic cells and quantification of the number of bacterial cells surviving the opsonic killing. As shown in Fig. 8B, while the pre-immune sera were completely ineffective, all three constructs induced antibodies with potent in vitro killing activity. mQβ–PNAG26 (EC50: 2534) and mQβ–PNAG10 (EC50: 3045) showed higher EC50 OPK titers as compared to mQβ–PNAG0

(EC50: 1345). Omitting either immune sera, complement or phagocytic cells resulted in complete loss of killing activity (Supplementary Fig. 7), indicating the need for all three components for protective immunity.

## Protection against *S. aureus* induced death by immunization

The efficacy of the various vaccine constructs in protecting against bacterial infection was tested in two mouse bacteremia challenge models. According to the CDC, bloodstream infections by *S. aureus* are serious threats with nearly 20,000 deaths per year in the USA[4]. For the in vitro study, we first compared the mQβ–PNAG0 vs TTHc–PNAG0 construct. In the active protection model, mice were immunized three times with mQβ–PNAG0 or TTHc–PNAG0 at equivalent doses (8 nmol PNAG0) (*n* = 20 for each group) (Fig. 9). Another group of control mice received a mock injection of saline. Two weeks following the last vaccination, each mouse was challenged via the tail vein with 10*LD50 of the *S. aureus* strain ATCC29213. Mice that had received saline all died within 2 days of bacterial challenge. On the other hand, 95% of the mice receiving mQβ–PNAG0 were protected against death from this pathogen. The survival rate of the mQβ vaccine group was significantly better than TTHc–PNAG0 vaccinated group (*p* = 0.0154, logrank test) (Fig. 9A). Bacteria were detected in the kidneys of 35% (7 out of 20) of the mice immunized with TT–PNAG0, while mQβ–PNAG0 vaccination reduced the recovered levels of *S. aureus* from the kidneys with bacteria only observed in 5% of the mice (1 out of 20) (Fig. 9B). Contingency table analysis of the proportion of the 20 immunized mice in each group with or without detectable *S. aureus* by Fisher's exact test showed significantly (*p* = 0.0436) fewer infected kidneys in the mQβ–PNAG0 immunized group, with a relative risk of 0.68 (95% CI = 0.45–0.93). Thus, disease burden evaluated by the levels of *S. aureus* in mouse kidneys was significantly better in mQβ–PNAG0 vaccinated group compared to those receiving the TTHc–PNAG0 vaccine. These results further support the superior performance of the mQβ carrier.

As the mQβ–PNAG0 immunogen gave almost complete protection in the active protection model in mice, we next established a passive protection model to differentiate the various mQβ–PNAG constructs, using rabbit sera transferred to mice. The passive model can be a more stringent test by using more dilute sera for protection.

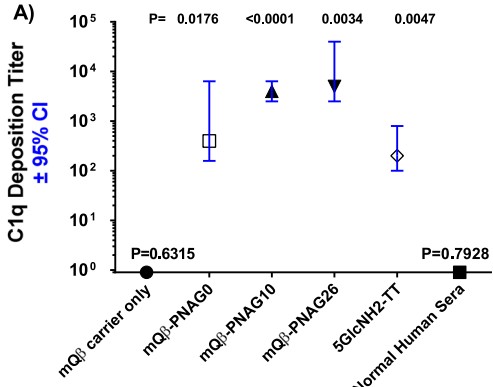
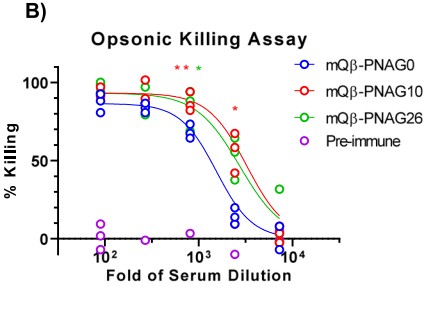
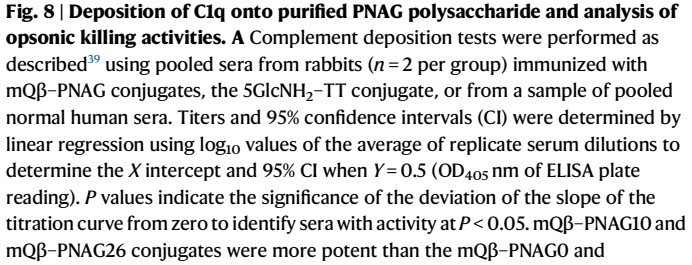

**Fig. 8 | Deposition of C1q onto purified PNAG polysaccharide and analysis of opsonic killing activities. A** Complement deposition tests were performed as described[39] using pooled sera from rabbits (*n* = 2 per group) immunized with mQβ–PNAG conjugates, the 5GlcNH₂–TT conjugate, or from a sample of pooled normal human sera. Titers and 95% confidence intervals (CI) were determined by linear regression using $\log_{10}$ values of the average of replicate serum dilutions to determine the *X* intercept and 95% CI when *Y* = 0.5 (OD₄₀₅ nm of ELISA plate reading). *P* values indicate the significance of the deviation of the slope of the titration curve from zero to identify sera with activity at *P* < 0.05. mQβ–PNAG10 and mQβ–PNAG26 conjugates were more potent than the mQβ–PNAG0 and

5GlcNH₂–TT conjugate in inducing C1q deposition onto purified PNAG. Normal human serum had no significant C1q depositing activity in spite of having a binding titer to PNAG (see Fig. 7B) consistent with prior reports that naturally acquired human antibody to PNAG is not functional due to the inability to activate the complement pathway[12,34]. Titers were determined by simple linear regression. **B** Pooled sera from rabbits (*n* = 2 per group) immunized with mQβ–PNAG conjugate led to significantly higher levels of opsonic killing activities against *S. aureus* cells. Three aliquots were prepared from each pooled serum and the individual values of the three aliquots were presented. Source data are provided as a Source Data file.

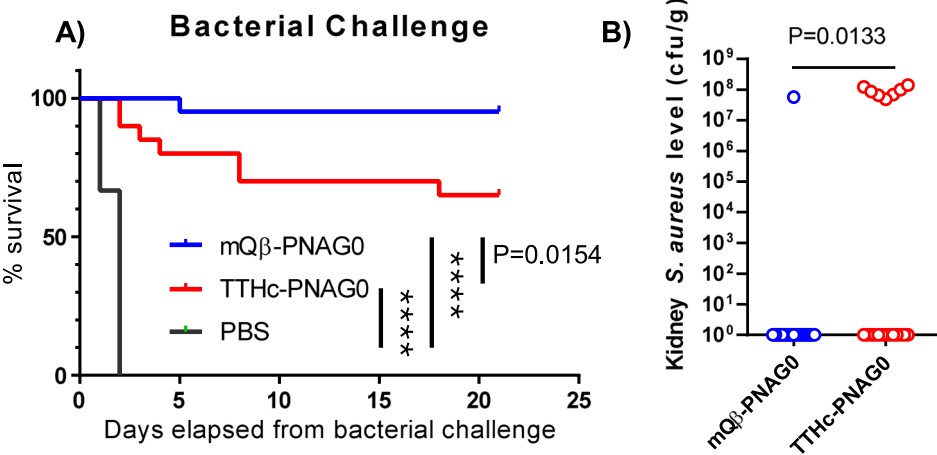

**Fig. 9 | Protective effects of vaccination.** Immunization with mQβ–PNAG0 effectively **A** protected against *S. aureus* infection, and **B** reduced bacterial count in mouse kidney. mQβ–PNAG0 was significantly better than TTHc–PNAG0 in protecting mice and reducing disease burden ($n = 20$ for each group). Logrank tests were performed for statistical analysis. *P* values were presented in the graph. ****$P < 0.0001$. Source data are provided as a Source Data file.

Rabbit sera were diluted 800-fold and administered intraperitoneally to mice, which were then challenged with 10*LD50 (200 million cells) of *S. aureus* ATCC29213 via the tail vein (Fig. 10). While all control mice receiving the pre-immune sera died within 3 days of this challenge, all post-immune sera from PNAG0, PNAG10, or PNAG26 immunized rabbits bestowed significant protection.

Mice receiving sera from mQβ–PNAG26 and mQβ–PNAG10 immunized rabbits showed higher survival rates than those receiving PNAG0 sera (Fig. 10A) (60% and 50%, respectively, vs 30%) and lower pathogen load compared to mQβ–PNAG0 sera supporting the in vitro opsonic killing data (Fig. 10B). We next tested the combination of two sera. Interestingly, administering the mixed PNAG26 and PNAG0 sera (1:1 ratio with each individual serum diluted 1600 times, which is regarded equivalent in concentration to 1:800 dilution of a single serum) provided 100% protection to mice against the 10*LD50 challenges with *S. aureus* (Fig. 10A). The kidneys of mice receiving the combination of PNAG26 and PNAG0 sera had no detectable bacteria (Fig. 10B). The higher protective efficacy observed with the combined sera was presumably because the PNAG polysaccharide can be heterogenous in the amine/acetylation patterns. While some of the sequences such as the fully deacetylated PNAG0 may be rare within the native PNAG polysaccharide, antibodies generated by mQβ–PNAG0 can complement those by mQβ–PNAG26. Thus, the combination of two mQβ–PNAG constructs can broaden bacterial recognition and enhance protection against bacterial challenges.

### mQβ–PNAG vaccines are effective against MRSA challenges

The emergence of MRSA is a pressing public health concern[40]. Effective vaccines can provide a complementary tool to combat *S. aureus* infections and reduce the reliance on antibiotics. The post-immune rabbit sera were tested against multiple MRSA strains including six clinical strains first via immunofluorescent staining (Supplementary Fig. 5B and Supplementary Table 1). All three mQβ–PNAG sera recognized the seven strains tested highlighting the breadth of immune recognition. A control strain lacking PNAG expression with *icaA* gene knock out (954) showed negligible binding by the immune sera, indicating the recognition is PNAG dependent. Next, rabbit sera were diluted 800 times and administered to mice, which were then challenged with 10*LD50 (200 million cells) of the MRSA strain 1058 via the tail vein (Fig. 10C). Sera from mQβ–PNAG26 immunized rabbits protected 90% of the mice from MRSA-induced death, which was significantly higher than the 40% protection by mQβ–PNAG0 sera.

Correspondingly, mice receiving mQβ–PNAG26 rabbit sera had the lowest overall bacterial load in the kidneys of challenged mice (Fig. 10D).

### Immunization does not significantly alter gut microbiome

As PNAG is expressed in many types of bacteria, we explored the effects of immunization on gut microbiome. To analyze the composition of the gut microbiome, mice were fully immunized with mQβ–PNAG26, and feces were collected on day 0 prior to immunization and day 35 following the initial prime immunization. The microbial species present in the droppings were analyzed via the 16S rRNA sequencing. Despite the significant amounts of anti-PNAG IgG produced in mouse sera, there were no significant changes in the microbial community present in the mouse gut (Supplementary Fig. 8). Similar results were reported in a sponsored trial of the 5GlcNH$_2$–TT vaccine and shown in the study of anti-PNAG therapy in the setting of graft-versus-host disease[41], or in human subjects in phase 1 clinical trials of both the 5GlcNH$_2$–TT vaccine or anti-PNAG mAb[42,43]. These observations corroborate that immunity to PNAG does not significantly alter the gut microbiome in immunized animals highlighting the potential safety of the vaccine.

In summary, numerous pathogens produce PNAG, rendering it a highly attractive target for vaccine development with the conjugate of fully deacetylated PNAG pentasaccharide with TT carrier currently undergoing human clinical trials as an anti-microbial vaccine. In order to enhance the potential protective efficacy, several aspects of the PNAG-based vaccine can be improved. As carbohydrates are typically T cell independent B cell antigens, an immunogenic carrier is critical. We have demonstrated that mQβ is a powerful carrier. The mQβ–PNAG conjugate was found to be superior in inducing higher levels of anti-PNAG IgG antibodies as compared to the corresponding PNAG conjugate with the TTHc carrier.

Besides the carrier, another important factor in vaccine design is the identification of the protective epitope(s) of the PNAG antigen, which was hampered by the lack of diverse structurally well-defined PNAG compounds. To gain a deeper understanding of the epitope specificity, a comprehensive library of PNAG pentasaccharides covering all possible combinations of free amine and NHAc has been synthesized. The synthesis is highlighted by a divergent design through the judicious choice of four amine protective groups, which can be orthogonally removed without affecting each other. The library of 32 PNAG pentasaccharides was obtained from just two strategically

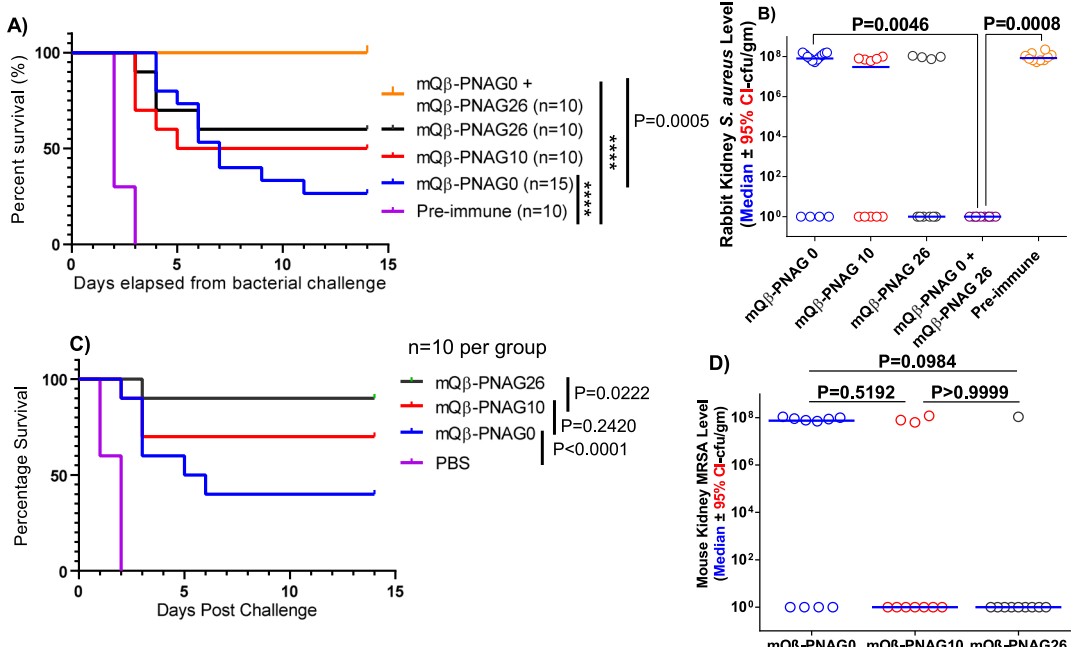

**Fig. 10 | Protective effects of antisera from immunized rabbits.** Transfer of antisera from mQβ–PNAG immunized rabbits to mice **A** provided significant protection to mice (*n* = 10 per group) against the lethal challenges by *S. aureus* ATCC29213. Statistical analysis was performed with the logrank test. ****$P < 0.0001$; and **B** significantly reduced bacterial count in mouse kidneys. The combination of sera from mQβ–PNAG0 and mQβ–PNAG26 immunized rabbits provided complete protection to mice. Statistical analysis for survival was performed using the logrank test. Analysis of *S. aureus* cfu/gm was by Kruskal–Wallis non-parametric ANOVA (*P* < 0.0001 for overall effect of serum given). *P* values for pairwise comparisons are shown on graph by Dunn's multiple comparisons test. Transfer of antisera from mQβ–PNAG immunized rabbits to mice **C** provided significant protection to mice against the lethal challenges by MRSA strain 1058 (*n* = 10 per group); statistical analysis for survival was performed using the logrank test; and **D** reduced bacterial count in mouse kidneys. Sera from mQβ–PNAG26 immunized rabbits provided the highest protection to mice. The horizontal line represents the median value of each group. Statistical analysis for survival was performed using the logrank test. Analysis of MRSA cfu/gm was by Kruskal–Wallis non-parametric ANOVA (*P* = 0.0967). *P* values for pairwise comparisons are shown on graph by Dunn's multiple comparisons test. Source data are provided as a Source Data file.

protected pentasaccharide intermediates, thus significantly enhancing the overall synthetic efficiency.

The availability of the comprehensive library provided an exciting opportunity to probe the epitope specificity through a glycan microarray. Screening of an anti-PNAG mAb F598 on the microarray suggests that the NHAc at unit B plays a critical role in F598 binding. NHAc at unit D could further enhance the binding. This knowledge led to the addition of two PNAG sequences (PNAG10 and PNAG26) beyond the fully deacetylated PNAG0 for vaccine studies.

The mQβ conjugates with PNAG10 and PNAG26 were found to elicit IgG antibodies capable of inducing high levels of complement deposition and opsonic killing of bacteria compared to the mQβ–PNAG0 conjugate. Vaccination with mQβ–PNAG conjugate provided effective protection to mice against lethal challenges by *S. aureus* in both active and passive immunity models. Mice were also effectively protected from MRSA-induced death by the immune sera with significantly reduced bacterial load in the kidneys. The vaccines are biocompatible with no adverse side effects and do not significantly disturb the gut microbiome of the immunized mice. PNAG-based vaccine design guided by the well-defined synthetic library of PNAG is a powerful strategy to develop the next generation of vaccines and more effectively fight against pathogen infections including those by drug resistant strains.

## Methods

### Animal study ethics statement

All animal care procedures and experimental protocols have been approved by the Institutional Animal Care and Use Committee of Michigan State University (protocol number: 202200444). All of the procedures described were performed in an AAALAC-accredited facility. Animals are housed with a 12:12 h light:dark cycle at a cage density of no more than 5 mice per cage. The housing rooms are maintained at 10–15 air changes per hour, room temperature maintained at 72 ± 2 °F, and relative humidity between 30 and 70%. Animals are housed within an individually ventilated caging system and on disposable bedding with enrichment. Wire-lid food hoppers within cages are filled with a standard rodent diet and animals are provided reverse-osmosis-purified water in bottles. The diet and water are available without restrictions.

### Data analysis software

All statistical analysis of data was performed using GraphPad Prism 6.

### General experimental procedures for synthesis and characterization

Anhydrous dichloromethane and diethyl ether were obtained from solvent purification columns. All reactions sensitive to moisture or oxygen were carried out under nitrogen in flame-dried or oven-dried (110 °C) glassware, which were cooled down to room temperature under vacuum or in a desiccator. A total of 4 Å molecular sieve powder was activated by flame-drying, cooled down under vacuum, and kept under nitrogen gas immediately prior to use in reactions. The progress of the reactions was monitored by thin layer chromatography (TLC) using glass TLC plates with compounds detected by UV light (254 nm) and/or by staining with 5% $H_2SO_4$ in ethanol or cerium molybdate stain ($Ce(NH_4)_2(NO_3)_6$ (0.5 g) and $(NH_4)_6Mo_7O_{24}\cdot 4H_2O$ (24.0 g) in 6% $H_2SO_4$ (500 mL)). Flash column chromatography purification was performed using silica gel P60 (230–400 Mesh). NMR spectra including $^1H$–NMR,

[1H]–[1H] gCOSY and/or [1H]–[13C] gHSQC spectra were acquired and referenced using CHCl$_3$ ($\delta$ [1H]−NMR 7.26 ppm) and CDCl$_3$ ($\delta$ [13C]-NMR 77.0 ppm).

## General procedure for pre-activation-based glycosylation

To a solution of donor (0.2 mmol) in DCM (2 mL) was added freshly activated 4 Å molecular sieve (100 mg). The mixture was stirred at room temperature for 30 min, and then cooled down to −78 °C. A solution of AgOTf (0.6 mmol) in Et$_2$O/DCM (-1:1 ratio, 0.5 mL) was added directly to the reaction mixture using a syringe without touching the flask wall. After stirring for 5 min, orange colored p-TolSCl (29 μL, 0.2 mmol, 1 eq to donor) was added to the reaction through a microsyringe. The yellow color should dissipate quickly within several seconds. The reaction was stirred until the complete activation of donor as determined by TLC analysis, which usually took <10 min. Then, a solution of acceptor (0.18 mmol, 0.9 eq) in DCM (0.5 mL) was added slowly to the reaction along the wall of the flask via a syringe. The reaction mixture was subsequently warmed to −20 °C in 1 h with continuous stirring. When the reaction was judged complete with the complete disappearance of the acceptor, the reaction was quenched with N,N-diisopropylethylamine (DIPEA, 100 μl) and filtered over Celite using a sintered filter funnel. The Celite layer was thoroughly washed with DCM until TLC showed the filtrate to be clear of any organic compounds. The filtrates were then combined, extracted with ethyl acetate, and washed with NaHCO$_3$ (sat. aq) and brine. The organic phase was dried over anhydrous Na$_2$SO$_4$ followed by filtration. The filtrate was concentrated under reduced pressure and purified by silica gel flash chromatography.

## General procedure for Alloc removal

To a solution of a pentasaccharide (6–50 mg scale) in DCM (2 mL) was added Pd(PPh$_3$)$_4$ (0.1 equiv) and phenylsilane (1 equiv). The reaction was stirred at room temperature for 3 h, after which it was directly loaded onto silica gel column (MeOH/DCM = 1/20) to afford the desired product.

## General procedure for Boc removal

The pentasaccharide was dissolved in TFA/H$_2$O (9/1, 6-50 mg compound in 2 ml) and the reaction was stirred at room temperature for 3 h. The solvent was removed. Then the mixture was loaded onto Sephadex LH-20 size exclusion chromatography (MeOH/DCM = 1/1) to afford the desired products.

## General procedure for Troc removal

To a solution of a pentasaccharide (6–50 mg scale) in THF and acetic acid (3:1, 1.2 mL) was added zinc dust (80 mg). The reaction was stirred at room temperature for 2 h, after which it was filtered through Celite, and the filtrate was concentrated under reduced pressure. The residue was purified by Sephadex LH-20 size exclusion chromatography (MeOH/DCM = 1/1).

## General procedure for Fmoc removal

The pentasaccharide was dissolved in piperidine/DMF (1/4, 6-50 mg compound in 1 ml) and the reaction was stirred at room temperature for 10 min. Then the mixture was loaded onto Sephadex LH-20 size exclusion chromatography (MeOH/DCM = 1/1) to afford the desired products.

## General procedure for acetylation

To a solution of a pentasaccharide (6–50 mg scale) in pyridine (0.8 mL) was added acetic anhydride (0.4 mL). The reaction was stirred at room temperature for 1 h, after which it was quenched by methanol and the mixture was concentrated under reduced pressure. The residue was purified by Sephadex LH-20 size exclusion chromatography (MeOH/DCM = 1/1).

## General procedure for removal of Bz and/or TFA

To a solution of a pentasaccharide (6 mg) in methanol (0.8 mL) was added hydrazine hydrate (0.2 mL). The reaction was stirred at room temperature for 8 h, after which it was diluted with methanol and quenched with 90% TFA to slightly acidic pH at 0 °C. The mixture was concentrated to 1 ml volume under reduced pressure. The residue was purified by Sephadex G-10 size exclusion chromatography (0.6% TFA in water).

## Synthesis of mQβ−PNAG conjugates

The mQβ particle (Qβ A38K/A40C/D102C)[26] was functionalized first with the SBAP linker. To a solution of 10 mg/mL mQβ in KPB (100 mM, pH = 7.0) was added 3.2 mg SBAP (15 equiv to mQβ subunits) in 10 μL DMSO. The solution was gently mixed and nutated under RT for 1 h. The protein was recovered with a 15 ml Amicon filter (MWCO = 100 kDa) under 4000 relative centrifugal force (rcf) and washed with KPB (100 mM, pH = 8.0) for 4 times. The modified mQβ was resuspended in KPB (100 mM, pH = 8.0) to a final concentration of 10 mg/mL and used immediately. Conjugation reactions of mQβ with PNAG0, PNAG10, and PNAG26 to mQβ were conducted in similar manners. As an example, PNAG0 (5 mg) was dissolved in 1 ml of degassed KPB (100 mM, pH = 8.0). Tris(2-carboxyethyl)phosphine, immobilized on Agarose CL-4B (Sigma-Aldrich # 52486, 2 equiv) was added and the solution was nutated under RT for 1 h to convert the disulfide bond in PNAG0 to free sulfhydryl groups. The resin was removed by centrifugation and the supernatant was lyophilized. To 5 mg of the reduced oligosaccharide (15 equiv to each mQβ subunit) was added 5 mg of SBAP functionalized mQβ in 0.5 ml KPB (100 mM, pH = 8.0) and the vial was gently inverted a few times to allow the oligosaccharide to dissolve. The mixture was nutated under room temperature for another 16 h before 5 μl 1 M cysteine was added. The mixture was nutated for another 2 h to quench the unconsumed bromoacetamide. The final product was recovered by a 15 mL Amicon filter (MWCO = 100 kDa) under 4000 rcf and washed with sterile PBS for 4 times. The final conjugate was analyzed by MALDI-TOF MS (Supplementary Fig. 2A).

## MALDI-TOF MS protocol

mQβ−PNAG conjugates were first reduced by DTT (100 mM) at 37 °C for 30 min to break up the disulfide bonds crosslinking mQβ subunits. BSA and TTHc conjugates were analyzed without pretreatment. Sinapinic acid (SA) was used as matrix. A total of 0.5 mg/ml SA was dissolved in acetonitrile:water:formic acid (50:50:0.1) mixture and 1 μL matrix solution was added to each spot of a MALDI base plate and air dried. The protein solution (1 μL, typical concentration: 1 mg/mL) was bound to a C4 resin tip (Millipore ZTC04S008) and desalted with water. The protein was then eluted with 1 μL acetonitrile:water:formic acid (50:50:0.1) mixture directly to the MALDI base plate to avoid protein loss on hydrophobic surfaces. The mixture was air dried and analyzed with MALDI-TOF MS.

## BSA conjugation

All BSA−PNAG conjugates were synthesized in the same manner. BSA powder (Sigma) was dissolved in potassium phosphate buffer (100 mM, pH = 7.0) to a concentration of 10.0 mg/mL. To 1 mL of BSA solution, 2.0 mg (42 equiv) of the SBAP linker (Fisher) was added in 10 μL DMSO. The solution was mixed gently on a nutator at RT for 2 h before passing through a 10 ml hand packed G25 column. The protein-containing fractions were recovered using Milli-Q water, aliquoted, and lyophilized. PNAG glycan (1 mg) was dissolved in a rubber stopper sealed vial with 0.5 mL degassed KPB (100 mM, pH = 8.5). 0.15 mg (1.0 equiv) of TCEP (sigma) was dissolved in 0.1 mL degassed KPB and injected into the vial with an HPLC syringe. The mixture in the vial was stirred under continuous N$_2$ flush for 30 min. The solution was injected into another sealed and N$_2$ flushed vial containing 1 mg lyophilized modified BSA. The vial was stirred for another 2 h. A cysteine

solution (1 M, 1 µl) was injected into the vial and the reaction was allowed to proceed for another 1 h to quench the unconsumed bromoacetamide. The protein was recovered by a G25 column and lyophilized. The saccharide loading was quantified by MALDI-TOF MS (Supplementary Fig. 3).

## Synthesis of TTHc–PNAG0

The buffer for recombinant TTHc (Fina Biosolutions) was first exchanged to KPB (100 mM, pH = 7.0) via a 15 ml Amicon filter (MWCO = 30 kDa) with repeated concentration and resuspension. The protein was then modified with SBAP. To 1 ml KPB solution of 5 mg/mL, TTHc was added 1.5 mg SBAP (50 equiv) in 10 µL DMSO. The solution was gently mixed and nutated under RT for 1 h. The protein was recovered with a 15 mL Amicon filter (MWCO = 30 kDa) under 4000 rcf and washed with KPB (100 mM, pH = 8.0) for 4 times. The modified TTHc was resuspended in KPB (100 mM, pH = 8.0) to a final concentration of 5 mg/ml and used immediately. To 4.6 mg of reduced oligosaccharide (50 equiv) was added 5 mg modified TTHc in 1 mL KPB (100 mM, pH = 8.0), and the vial was inverted gently to allow the oligosaccharide to dissolve. The mixture was nutated under RT for another 16 h before 10 µL 1 M cysteine was added. The mixture was nutated for another 2 h to quench the unconsumed bromoacetamide. The final product was recovered by a 15 mL Amicon filter (MWCO = 30 kDa) under 4000 rcf and washed with sterile PBS 4 times. The final conjugate was analyzed by MALDI-TOF MS (Supplementary Fig. 2B).

## Glycan microarray

Glycan microarray slides were produced as previously described[44,45] on SuperEpoxy 2 slides (SME2; ArrayIt Corp, Sunnyvale, CA) and stored vacuum sealed at −20 °C. Prior to use, slides were warmed to RT and then scanned to identify any missing spots or defects (the print buffer contains a soluble dye). Slides were printed with 8 replicate array blocks. An 8-well slide module (ProPlate® Multi-Well Chambers, Grace Bio-Labs) was affixed to each slide to separate the 8 blocks. Each of the 8 wells was then blocked overnight at 4 °C with 400 µL/well of 3% w/v BSA (Sigma-Aldrich, A3059, Lot SLBW6770) in PBS. Next, slides were washed twice with 400 µL/well of PBST followed by incubation with rabbit serum. Serum samples were diluted into 1% BSA/PBS (1:50,000, 1:500,000, and 1:5,000,000 dilutions for rabbit sera) and then incubated at 37 °C with gentle shaking for 2.5 h. Unbound sera were removed from each well using a multichannel pipettor, followed by three quick washes with 400 µL/well of PBST and three 2 min washes with PBST. Wells were next incubated with fluorophore-labeled secondary reagents [Cy3 anti-rabbit IgG (Cy™3 AffiniPure Goat Anti-Rabbit IgG, Fc fragment specific; Jackson 111-165-046) and AlexaFluor 647 Anti-rabbit IgM (Alexa Fluor® 647labeled Goat Anti-Rabbit IgM mu chain; Abcam 150095), each diluted 1:500 in 1% BSA/PBS] for 45 min at 37 °C with gentle shaking. Secondary reagent solutions were removed, followed by four quick washes with 400 µL/well of PBST and one 2 min wash with PBST. The well module was removed, and the slide was submerged in PBST for 5 min as a final wash. Slides were dried by centrifugation at 1000 × g for 4 min and then scanned on an InnoScan 1100 AL fluorescence scanner (Innopsys; Chicago, IL) at 5 µm resolution. Analysis was performed using GenePix Pro 7 software (Molecular Devices Corporation; Sunnyvale, CA). Background fluorescence was subtracted from the median spot fluorescence and values were averaged for each duplicate component spot. Additional details on array fabrication, processing, and analysis can be found in the protocol by Temme & Gildersleeve[46]. Full microarray data can be found in the Source Data file for Fig. 7C.

## Immunization protocol

Vaccination constructs were prepared by mixing the conjugates in sterile PBS and MPLA. The dosing of the constructs was determined by average antigen loading on the carrier to allow 8 nmol glycan dose for each mouse. In a typical protocol, 100 µg mQβ–PNAG0 in 180 µl PBS was mixed with 20 µg MPLA in 20 µl DMSO to produce one dose of the vaccine.

Pathogen-free C57BL/6 or CD1 female mice aged 6 weeks were obtained from Charles River and maintained in the University Laboratory Animal Resources facility of Michigan State University. Only female mice were used due to the ease of housing as male mice initially cohoused often must be separated or euthanized due to fighting, so many are lost from the study group. Groups of five mice were injected subcutaneously under the scruff with the MPLA adjuvanted vaccine constructs. Immunization procedures were performed on day 0, 14, and 28. Blood samples were collected via the hind limb vein on days 0, 7, 21, and 35.

Rabbit immunization was performed by ProSci. Female New Zealand white rabbits (n = 2 per group, aged 15 weeks) were immunized with the mQβ–PNAG vaccine construct with each dose containing an equivalent glycan (PNAG0, PNAG10, or PNAG26) amount of 8 nmol. The prime vaccination was adjuvanted with the Complete Freund's Adjuvant (CFA) and the booster doses were adjuvanted with the incomplete Freund's Adjuvant (IFA). For each dose, 100 µl vaccine conjugate solution was mixed with 100 µl CFA or IFA and vortexed vigorously to form a homogeneous emulsion. The immunization procedures were performed on days 0, 14, 28, and 42. Blood samples were collected on days 0, 7, 21, and 35. The terminal bleeding was performed on day 56.

## ELISA assays to determine antibody binding to oligosaccharides conjugated to BSA

Immulon 4 HBX 384 well plates (ThermoFisher 8755) were coated with 0.5 µg BSA–PNAG conjugates in PBS (50 µL) at 4 °C overnight. The liquid was discarded, and the wells were blocked by 100 µL PBS containing 1% BSA for 1 h. The liquid was discarded, and the plates were washed by PBST (0.05 v/v Tween 20 in PBS) for 4 times. Mouse or rabbit immune sera were serial diluted in PBS containing 0.1% BSA. Diluted serum (50 µL) was added to each well and incubated for 2 h. The liquid was discarded, and the plates were washed with PBST for 4 times. Peroxidase AffiniPure-goat anti mouse IgG antibody (Jackson 115-035-062, 1:2000 dilution) in PBS containing 0.1% BSA (50 µL) was added and incubated for 1 h. For rabbit serum, the peroxidase AffiniPure-goat anti-rabbit IgG antibody was used (Jackson 111-035-045). The liquid was discarded, and the plates were washed with PBST for 4 times. TMB substrate solution (75 µL) was added to each well and the reaction was allowed to proceed for 15 min before 25 µl 0.5 M sulfuric acid was added. $OD_{450nm}$ was immediately recorded and fitted into the 4PL non-linear logistic model via GraphPad Prism 6 with least squares algorism. Maximum OD values reached ~3.0 for all groups. Unmodified BSA-coated wells showed negligible signal (<0.1 AU) and were subtracted as plate blank. EC50 titers were calculated to present the dilution folds where binding signal is 50% of the maximum: the average value (OD-1.5) between the maximum and baseline was used as intercept to calculate EC50 titers. All dilutions were run as triplicates. All steps were run under ambient temperature unless specified.

## ELISA assays to determine binding to native PNAG polysaccharide and analysis of deposition of complement component C1q

ELISA assays to determine antibody titers to native PNAG polysaccharide and analysis of the deposition of complement component C1q were performed as described[39] except that mouse or rabbit antisera and the corresponding secondary antibodies were used. Native PNAG polysaccharide was purified from *A. baumannii* bacteria[37,38]. Immulon 4 HBX 96-well plates (ThermoFisher 8755) were coated with purified PNAG polysaccharide (100 µL/well, 0.6 µg/mL PNAG in 40 mM phosphate buffer, pH 7.2) overnight at 4 °C. The plates were treated with cold methanol (−20 °C, 100 µL) for 5 min before being blocked for

60 min at 37 °C with 1% (w/v) skim milk (pasteurized at 65 °C for 60 min) in PBS (120 μL). The plates were washed three times with PBS with 0.05% Tween 20 (PBST; 200 μL each time). Sera were added to a solution of PBST containing 1% skim milk (1 g/100 mL). Serial dilutions of serum solutions were added to the ELISA wells (100 μl/well, starting at 1:100 dilution for all samples with twofold dilutions down). The plates were incubated for 1 h at 37 °C and washed three times with PBST (200 μL each time). Alkaline phosphatase-conjugated goat anti mouse IgG antibody (Sigma-Aldrich AP308A, 1:2000 dilution) or goat anti-rabbit IgG antibody (Sigma-Aldrich AP307A) in PBS containing 1% skim milk (100 μL) was added and incubated for 1 h at 37 °C. The liquid was discarded, and the plates were washed with PBST 3 times. Di-sodium 4-nitrophenyl phosphate (1 mg/mL, 100 μL) in substrate buffer (1.6 g/L $NaHCO_3$, 2.92 g/L $Na_2CO_3$, and 100 mg/L $MgCl_2$) was added to each well and the AP-mediated color development was allowed to proceed for 30 min at 37 °C. $OD_{405}$ nm for each plate was recorded immediately. Titers were calculated by simple linear regression as the values (fold dilution on the $X$ axis) where the curve crossed the $Y$ axis at the OD reading determined by adding 3 times the standard deviation to the mean upper OD limits of control wells with pre-immune sera.

The ability of each test serum to deposit C1 onto PNAG antigen was tested in quadruplicates in four parallel wells. Samples of each serum were heated at 56 °C for 30 min to inactivate the endogenous complement in the serum. This is important because the endogenous complement within each sample may vary. Heat-inactivation of endogenous complement removes this variable. To each individual well was added each serum at an appropriate dilution (50 μL) made in gelatin-veronal buffer (GVB, Boston BioProducts IBB-300X) with a complement source added (2.5% PelFreeze IgG/IgM Depleted Human Complement Pooled Serum diluted to 2.5% in GVB; 50 μL). The plates were incubated for 1 h on a rocker at 37 °C, and washed 3 times with PBST. Subsequently, affinity-purified goat anti-human complement C1q antibody (Cedarlane, Inc, CL7341AP, 100 μL, 1:1000 dilution in GVB) was added to each well and incubated for 1 h under ambient temperature. This was washed three times with PBST, after which anti-goat IgG-alkaline phosphatase secondary antibody (100 μL, 1:2000 dilution in GVB) was added. Plates were incubated for 1 h at ambient temperature. After the plates were washed 3 times, di-sodium 4-nitrophenyl phosphate (1 mg/mL, 100 μL) in substrate buffer (1.6 g/L $NaHCO_3$, 2.92 g/L $Na_2CO_3$, and 100 mg/L $MgCl_2$) was added to each well and the alkaline phosphatase-mediated color development was allowed to proceed for 30 min at 37 °C. $OD_{405}$ nm for each plate was measured in a 96-well plate reader. Titers were calculated by simple linear regression as the $OD_{405}$ nm values (fold of dilution on the $X$ axis) where the curve crossed the $Y$ axis at an OD reading determined by adding 3 times the standard deviation to the mean upper OD limits of control wells with pre-immune sera. The rabbit antisera to the TT conjugate with $5GlcNH_2$ ($5GlcNH_2$−TT) were prepared as previously described[7,13].

## OPK killing assay
OPK killing assay was adapted from previously published work[47]. HL-60 (ATCC CCL-240) cells were maintained in RPMI-1640 with L-gluta-mine and 10% FBS (growth medium), at 37 °C and 5% $CO_2$ in sterile filter-capped 75 $cm^2$ flasks. Upon use, the cells were induced to differentiate with 0.6% (V/V) DMF added to growth medium. The induction was performed for 5 days. *S. aureus* (strain ATCC29213) was grown in LB medium to an OD of 0.7. Aliquots of the bacterium stock were kept at −80 °C in 1:1 (V:V) glycerol and LB. Bacterial concentrations in cfu/mL were determined by serial dilution and overnight growth on trypticase soy agar plate under 37 °C. The numbers of colonies were used to calculate cell density in the stock. Heat treated (56 °C, 30 min) immune sera were serial diluted in growth medium and mixed with bacterial cells suspended in the same medium in a 96-well plate. For each dilution, triplicated wells were used. Each well consists of

2000 cfu bacterial cells in 3 μL medium and 7 μL diluted immune serum (a 1:1 mixture of sera from two rabbits receiving the same immunization). The mixture was incubated under 37 °C for 30 min. Induced HL-60 cells were harvested and resuspended to a density of $1.25 \times 10^7$/mL in growth medium. To each well of bacterium-serum mixture was added 40 μL HL-60 cells and 10 μL baby rabbit complement (Cedarlane CL3442-R). The mixture was incubated under 37 °C with 5% $CO_2$ for 1 h, when 100 μL growth medium was added to each well. The mixture (10 μL) was spread on trypticase soy agar plates and the remaining live bacteria were counted by colonies formed after overnight incubation. Pooled sera (50:50 mixed sera from two rabbits receiving the same vaccine) were used for the study. HL-60 omitted groups, complement deactivated groups (56 °C, 30 min), pre-immune serum groups, and free bacterium groups were each performed as controls. Free bacterial groups were used as the standard for 0% killing. Bacterial omitted group was used as the standard for 100% killing. The killing % of experimental groups was plotted against final serum dilution and fitted to 4PX non-linear logistic model with least squares algorism (GraphPad Prism 6). The dilution that gave 50% killing for each serum was calculated and used as EC50 titer. Three aliquots of each pooled serum were prepared and the opsonic killing activities for each aliquot were determined and plotted in Fig. 8B.

## Immunofluorescent staining
Single colony of *S. aureus* strain ATCC29213 or MRSA strains grown on a trypticase soy agar plate were inoculated and cultured in LB medium to an OD of 0.7. Cells were then spread to a glass slide with a sterile cotton tip. The slides were air-dried and fixed in ice-cold methanol for 2 min. The methanol was allowed to air dry, and the slides were washed in PBST for 3 times. *S. aureus* endogenous antibody binding proteins such as protein G were then blocked with 20 μg/ml recombinant human Fc (Invitrogen A42561) in PBS with 1% BSA for 1 h under ambient temperature in a moisturized environment. The blocking solution was discarded, and the immune serum diluted in PBS with 1% BSA (1:1600) was added to the cells. The slides were kept under ambient temperature for 1 h in a moisturized environment and washed with PBST 3 times. Alexa Fluor 594 labeled goat anti-rabbit IgG Fc 2nd antibody (ThermoFisher A11012) was diluted to a concentration of 5 μg/ml and added to the cells. The staining was performed overnight under 4 °C in a dark and moisturized environment. The slides were then washed in PBST 3 times and sealed with curing anti-fade mountant for 24 h under ambient temperature. The slides were visualized under a Nikon A1 CLSM.

## Mouse challenge study protocol
For active immunization and challenge studies, female CD1 mice aged 5–6 weeks were grouped ($n = 20$) allowing a similar body weight average and distribution across the groups. Mice were immunized by TT–PNAG0 or mQβ–PNAG0 constructs each containing 8 nmol of oligosaccharides, adjuvanted with 20 μg MPLA. Twenty-one and Forty-two days post-priming, mice were boosted with the same constructs. Another group of mice received PBS as a control. Fifty-six days post-priming, $2 \times 10^8$ cfu *S. aureus* ATCC29213 cells in 200 μL PBS were intravenously administrated for each mouse. Mouse condition was monitored 3 times a day for the following 2 weeks. Mice losing more than 20% body weight or becoming moribund were euthanized. The kidneys were collected and homogenized, and bacteria loading was determined via serial dilution and growing on TSB agar plates. On 14 days post-challenge, all surviving mice were euthanized, and kidney bacterial levels were determined.

For passive immunization and challenge studies, female CD1 mice aged 12–13 weeks were grouped ($n = 10$) allowing a similar body weight average and distribution across the groups. Pooled ($n = 2$) rabbit sera collected on 56 days post priming from mQβ–PNAG 0, 10, or 26 groups were heat inactivated (56 °C, 30 min) and diluted 1:800 fold. Mice

intraperitoneally injected with 200 µL of diluted sera and challenged with $2 \times 10^8$ cfu *S. aureus* ATCC29213 cells in 200 µL PBS via the tail vein. Survival and disease burden of the challenged mice were monitored following the same protocol as described for active immunization and challenge studies.

**DNA extraction for microbiome analysis.** For the microbiome sequencing experiments, pathogen-free CD1 female mice were obtained from Charles River and housed in the University Laboratory Animal Resources facility of Michigan State University. Eight-week-old mice were injected subcutaneously under the stuff. The treatment group was injected with 0.1 ml MPLA adjuvanted vaccine constructs on days 0, 14, and 28. The control treatment was injected with 0.1 mL of 0.01 M DPBS instead. Stools were collected from vaccine- and control-treated animals at pre- and 42-day post-vaccination. DNA was extracted from feces using the QIAamp® PowerFaecal® Pro DNA kit (QIAgen®). The quality and purity of the isolated genomic DNA were confirmed by gel electrophoresis and quantitated with a Qubit2.0 fluorometer using the Qubit® dsDNA HS Assay Kits (Thermo Fisher Scientific). The DNA samples were stored at −20 °C.

### PCR amplification procedure
Amplification of the 16S rRNA gene (V3–V4) was performed using V3 (5′-CCT TACGGGAGGCAGCAG-3′), V4 (5′-GGA CTACHV GGG TWTCTAAT-3′) primers (W: A or T; V: G or C or A; H: A or C or T). The primers were custom synthesized by Integrated DNA Technologies. The PCR mixture for every reaction tube was composed of 12.5 µL DreamTaq™ Hot Start PCR Master Mix (Thermo Fisher Scientific) at a 2× concentration, along with 11.5 µL nuclease-free water and 0.5 µL of each primer (V3 forward primer and V4 reverse primer). 1 µL (10 ng/µL) template DNA was added to the 25 µl mastermix. Thermocycling conditions included an initial denaturation step (3 min at 94 °C), followed by 30 cycles of denaturation (30 s at 94 °C), annealing (30 s at 56 °C), extension (5 min at 72 °C), and a final extension step of 1 min at 72 °C. PCR products were separated using gel electrophoresis on 1.5% agarose gel.

### Library preparation and sequencing
The V3–V4 region of the 16S rRNA gene was amplified using indexed primers (341f/806r) suitable for the Illumina platform (Caporaso 2011). Libraries were normalized using Invitrogen SequalPrep DNA normalization plates and pooled products were cleaned and concentrated with Amicon DNA Concentrator and quantified with Qubit dsDNA HS, Agilent 4200 TapeStation HS DNA1000 and Kapa Illumina Library Quantification qPCR assays. The pooled libraries were run on a MiSeq platform using Illumina MiSeq v2 500 cycle chemistry in a standard V2 flow cell. Base calling was performed by Illumina Real Time Analysis (v1.18.54).

### Bioinformatics and statistics
The output MiSeq FastQ files were taken directly into Mothur v1.44.1[48] for phylogenetic analysis using the Silva database (V.132) for alignment and Training Set 18 from the Ribosomal Database Project for taxonomic identification. The command sequence of The Mothur MiSeq SOP (https://mothur.org/wiki/miseq_sop/) was used for the analysis with the exception that all samples were subsampled down to 30,000 sequences before the alignment step. As recommended in the MiSeq SOP, we allowed 1 base difference per 100 bp in the pre-cluster command. In addition, prior to the final analytical steps in Mothur, all samples were rarefied to 21,900 sequences. Both operational taxonomic unit (97%) and Amplicon sequence variants datasets were generated in Mothur. The statistical analyses were conducted in PAST 4.0 (Paleontological Statistics Software Package)[49] and based on 5507 OTUs after the removal of all singletons. A Bray-Curtis dissimilarity matrix was used to compute a Non-Metric Multidimensional

Scaling plot displaying the community-level differences amongst samples as well as computing pairwise comparisons using ANalysis Of SIMilarity and Permutational multivariate analysis of variance, all as implemented in PAST. To identify populations that contributed to community differences, SIMPER as implemented in PAST and LefSe through the Huttenhower LefSe website (https://huttenhower.sph.harvard.edu/lefse) were used.

The sequencing data have been submitted to Genbank/SRA (Bioproject ID PRJNA997319).

## Statistical analysis
The values for N, P, and the specific statistical test performed for each experiment are included in the appropriate figure legend or main text.

## Supporting information
The Supporting Information includes detailed experimental procedures, characterization data, glycan microarray data, and supplementary figures.

## Reporting summary
Further information on research design is available in the Nature Portfolio Reporting Summary linked to this article.

## Data availability
All data generated or analyzed during this study are included in this published article and its Supplementary Information files. The sequencing data are available from Genbank/SRA (Bioproject ID PRJNA997319). All other data are provided in the Supporting Information. Source Data are provided with this paper.

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

## Acknowledgements

We are grateful to the National Institutes of Health (Grant R01AI146210 (X.H.) and the National Cancer Institute Intramural Research Program (J.C.G.)), Michigan State University Foundation (X.H.), and the Hangzhou Leading Innovation and Entrepreneurship Team Project TD2022002 (W.Y.) for financial support of our work. We thank the Consortium for Functional Glycomics (GM62116; the Scripps Research Institute), T. Tolbert (University of Kansas), L.-X. Wang (University of Maryland), J. Barchi (National Cancer Institute), T. Lowary (University of Alberta), Omicron Biochemicals Inc., GlycoHub, and Glycan Therapeutics for generously contributing glycans for the array. The icaA::Tn strain (NE37) used in this study was isolated from the defined transposon mutant library and provided by the Network on Antimicrobial Resistance in *Staphylococcus aureus* (NARSA) for distribution by BEI Resources, NIAID, NIH: Nebraska Transposon Mutant Library (NTML) Screening Array NR-48501.

## Author contributions

Conceptualization: X.H. Methodology: Z.T., W.Y., T.M., N.H., G.B.P., J.C.G., X.H. Investigation: Z.T., W.Y., N.A.O., X.P., S.R., T.M., C.C.B., M.V. Supervision: T.M., G.B.P., J.C.G., X.H. Writing—original draft: Z.T., W.Y., X.H. Writing—review & editing: Z.T., W.Y., T.M., N.H., G.B.P., J.C.G., X.H.

## Competing interests

X.H. is the founder of Iaso Therapeutics Inc., which is dedicated to the development of next generation of vaccines. G.B. P. has a financial interest in Alopexx, Inc. a company developing broad-spectrum immune therapeutics, which target PNAG for the prevention, treatment, and mitigation of bacterial, fungal, and parasitic infections. G.B.P.'s interests were reviewed and are managed by Brigham and Women's Hospital and Mass General Brigham in accordance with their conflict-of-interest policies. All other authors declare no competing interests.
