## [Peer Review File · Nature Communications]

A comprehensive synthetic library of poly-N-acetyl glucosamines enabled vaccine against lethal challenges of *Staphylococcus aureus*REVIEWER COMMENTS

Reviewer #1 (Remarks to the Author):

The manuscript submitted to Nature Communications by Tan et al. describes the synthesis of a library of 32 PNAG-related pentasaccharides containing all possible acetylation patterns at the 2-amino functions of the D-glucosamine residues. PNAG is a well-known and key virulence factor expressed on many pathogens and component of the biofilm, in particular in various antimicrobial resistant strains such as *Staphylococcus aureus* and *Acinetobacter baumannii*, therefore it is an attractive vaccine target. The reported strategy appears robust. All key glycosylations are highly stereoselective and provide good yield (69-83%). The divergent approach enabled to obtain all library components leveraging in a smart way the orthogonality of different N-protecting groups. Although the synthetic route does not contain significant innovations, it is well-designed and well-executed, and the spectroscopic and analytical data of all new compounds are fully consistent with the proposed structures. My only remark for this section of the manuscript is at line 165, in which «by deacylation of the thioester» should be replaced by «by full O- and S-deacylation».

In the second part of the manuscript the authors report the chemical conjugation of the PNAG pentasaccharides with the mutant bacteriophage Qbeta and the immunological evaluation of the neoglycoconjugates obtained, using the corresponding conjugates with a variant of the widely employed TT carrier (namely, PNAG-TTHc conjugates) as a benchmark. Interestingly, Qbeta-conjugates raised high IgG titres in mice (higher than the corresponding TTHc-conjugates), and high antibody levels remained after one year post-immunization. Even more interesting, the IgG levels could be boosted to the initial titres after nearly two years, a hallmark of B cell-mediated immunological memory. Glycan microarray experiments using a monoclonal antibody allowed to map the NHAc vs NH₂ preference at each monosaccharide residue of the library components. Results showed that N-acetylation at residues B and D of each PNAG pentasaccharide provided the highest binding with the monoclonal antibody, and compounds designated as PNAG10 (diacetylated), PNAG26 (triacetylated) and PNAG30 (tetracetylated) were the strongest binders, along with PNAG31 (fully N-acetylated, however this compound had been found poorly immunogenic in previous reports). However the sentence at lines 255-256 («glycan microarray showed that F598 actually preferred highly acetylated PNAG with the PNAG30 and PNAG31...») is

misleading, since, according to Fig. 3A, PNAG26 shows higher Ab binding than PNAG31. The immunization protocol was repeated on rabbits, confirming the results obtained on mice. In a further experiment, the authors evaluated the ability of sera from rabbits immunized with Qbeta-conjugates to recognize native PNAG extracted from bacteria. All sera showed strong binding (the PNAG26 was the best), much higher than the binding observed with the sera from TT-conjugate, demonstrating again the superior performance of the Qbeta carrier. For clarity reason, part C and part D of Fig. 2 should be included in a different figure and moved ahead, closer to the text that describes them. The functional activity of the post-immune sera was evaluated by antibody-mediated complement deposition and opsonophagocytic killing assay.

Finally, both active and passive immunization experiments showed that the Qbeta-conjugates are able to provide effective protection of mice against lethal challenges by *S. aureus*. Particularly intriguing is the result described at lines 438-446: a 1:1 mixture of PNAG26 (triacetylated) and PNAG0 (non acetylated) sera provided 100% protection to mice against lethal challenges. This result seems to go against the hypothesis of the occurrence of a specific glycoepitope in the synthetic PNAG pentasaccharides. The authors should comment about this and try to provide a plausible interpretation of this result. For the same reason, I also suggest the authors to temper their claim of the introduction (lines 85-87), since no specific PNAG epitopes could be identified in this work.

As a general comment, in my opinion the manuscript is nicely written, well-organised, clear and easily understandable even for readers non expert in the field. The text is fluent and not verbose.

The list of references is exhaustive, in each section the most relevant previous works are appropriately cited.

There is no doubt that the library of synthetic PNAG pentasaccharides is a very valuable tool to guide the identification of protective glycoepitope(s), a key issue in the glycoconjugate vaccines development. However, the disclosed results do not show a clearly defined glycoepitope. In addition, it is unclear whether the superior immunogenicity of the novel neoglycoconjugates is due to the carrier Qbeta only, or (at least in part) to the precise acetylation pattern of the synthetic oligosaccharides. There are no convincing data that allow to dissect the role of the two components. It is well-known that the carrier can have a crucial role in determining the immunogenicity of the glycan payload, for example by

exposing the carbohydrate antigens in a better way to bind B cell receptors. The authors should take in account and discuss these aspects both in the introduction and, in particular, in the conclusion. Another issue concerns the carbohydrate loading in the conjugates. According to Fig. S1, the carbohydrate loading in all Qbeta-PNAG conjugates is the same (250 glycans per Qbeta particle). However, in the head-to-head comparison (lines 199-202) equal amounts of Qbeta-PNAG0 and TTHc-PNAG0 conjugates were administered: this means that the two conjugates have the same carbohydrate density? Can the authors comment about this point? More in general, can the authors clarify the role of the carbohydrate loading in their immunogenicity studies?

In conclusion, in my opinion this is a beautiful piece of work which deserves publication on Nature Communications with the revisions mentioned above.

Additional minor remarks are the following:

- line 134: it should be "the glycosylation of acceptor 8 with disaccharide 9..."
- Scheme 3, part B. cysteine instead of cycteine
- Figure 4B: correct with mQbeta(symbol)-PNAG (Y-axis) In addition, a) and b) in the caption should be replaced with A) and B)
- lines 406, 407, 413: I think it should be TTHc conjugates
- line 476: which mQbeta conjugate was used in this experiment?

Reviewer #2 (Remarks to the Author):

In their paper entitled "A comprehensive synthetic library of poly-N-acetyl glucosamines enabled vaccine against lethal challenges of *Staphylococcus aureus*" the authors describe an elegant synthesis of a panel of 32 PNAG pentasaccharide analogs with chemically defined N-acetylation patterns derived from a common set of two orthogonally protected pentasaccharide precursors. These chemically defined PNAG pentasaccharide analogs were then used as part of a glycan microarray to profile the binding specificity of a human anti-PNAG monoclonal antibody F598. The top hits were then conjugated to mutant bacteriophage Q β and used as vaccine candidates for immunization trials in mice and rabbits. The top performing PNAG-mQ β vaccine candidates performed better than a fully de-N-acetylated PNAG pentassaccharide-tetnus toxoid conjugate currently in human clinical

trials.

Overall, the manuscript is clear, and the results of this study will be of broad interest to the audience of Nature Communications. However, the authors should address the following concerns before the manuscript is considered for final publication:

The results presented in Figure 2 for immunization studies are discussed in two separate parts of the manuscript (on page 10 and then on page 14). To avoid discussing these same findings in two separate parts of the manuscript, this reviewer recommends that the authors move the discussion profiling the PNAG specificity of F598 to appear before the discussion of immunization trials reported currently in Figure 2.

Figure 2D. Should be plotted either showing the individual data points or as a box and whisker plot and should indicate the number of samples represented in each data point. The figure legend for figure 2D should also indicate the bacterial source of the native PNAG used in this study.

Figure S8 should appear before figures S6 or S7 in the supplemental information as it is mentioned in the text first. Also, the authors should include the statistical method used to determine significance in these comparisons.

The details for the method used to determine complement deposition (results are reported in Figure 5A) are not described in the manuscript. This method should be described in the materials and methods section in detail. The authors should also clearly indicate what statistical test is being used to determine statistical significance for the data reported in Figure 5A as it is not clear what comparisons the reported P-values refer to. The authors should report individual datapoints for the data shown in this graph.

For the opsonic killing assay results presented in Figure 5B and Figure S6, there appear to be three data points reported for each serum dilution, but according to the figure legend sera from n=2 rabbits were used. The methods also report that duplicate measurements for each serum dilution were measured. Thus, it is not clear what the data points reported in this

figure represent and why there are three data points at each serum dilution. The authors should clearly indicate what each data point represents (ie. are these the mean % killing values from duplicate wells, etc...).

Figure 7A/B. The authors need to indicate the number of mice used in each treatment group. This is indicated in Figure 7C/D but not in A/B.

In Figure 7D the t-test is not the appropriate statistical test to evaluate this data. Instead, a one-way ANOVA should be used, or since the distribution of the data appears to be bimodal (ie. there appears to be two distinct populations in each data set) a more appropriate analysis may be a Kruskal-Wallis non-parametric test.

A table detailing the bacterial strains used in this study and their sources should be include in the supplemental information of this article.

Figure S5. The authors should provide a more descriptive figure caption to clearly indicate the results observed in these studies. Bright field or phase contrast images for the fields of view presented in Figure S5 should also be provided to demonstrate that the samples contain similar amounts of bacterial cells.

Figure S7. The authors should fully write out all of the abbreviations used in the figure caption.

¹³C NMR data are reported for PNAG0-PNAG32 but the spectra are not provided as part of the supplemental information. The authors should provide these spectra in the SI if available.

Responses to Reviewers

Reviewer 1

1. **Comment 1:** The manuscript submitted to Nature Communications by Tan et al. describes the synthesis of a library of 32 PNAG-related pentasaccharides containing all possible acetylation patterns at the 2-amino functions of the D-glucosamine residues. PNAG is a well-known and key virulence factor expressed on many pathogens and component of the biofilm, in particular in various antimicrobial resistant strains such as *Staphylococcus aureus* and *Acinetobacter baumannii*, therefore it is an attractive vaccine target. The reported strategy appears robust. All key glycosylations are highly stereoselective and provide good yield (69-83%). The divergent approach enabled to obtain all library components leveraging in a smart way the orthogonality of different N-protecting groups. Although the synthetic route does not contain significant innovations, it is well-designed and well-executed, and the spectroscopic and analytical data of all new compounds are fully consistent with the proposed structures. My only remark for this section of the manuscript is at line 165, in which «by diacylation of the thioester» should be replaced by «by full O- and S-deacylation».

:

Response: We would like to thank the reviewer for the positive overall evaluation. At line 166, “by diacylation of the thioester” was revised to “by full O- and S-deacylation” as suggested.

2. **Comment 2:** In the second part of the manuscript the authors report the chemical conjugation of the PNAG pentasaccharides with the mutant bacteriophage Qbeta and the immunological evaluation of the neoglycoconjugates obtained, using the corresponding conjugates with a variant of the widely employed TT carrier (namely, PNAG-TTHc conjugates) as a benchmark. Interestingly, Qbeta-conjugates raised high IgG titres in mice (higher than the corresponding TTHc-conjugates), and high antibody levels remained after one year post-immunization. Even more interesting, the IgG levels could be boosted to the initial titres after nearly two years, a hallmark of B cell-mediated immunological memory. Glycan microarray experiments using a monoclonal antibody allowed to map the NHAc vs NH₂ preference at each monosaccharide residue of the library components. Results showed that N-acetylation at residues B and D of each PNAG pentasaccharide provided the highest binding with the monoclonal antibody, and compounds designated as PNAG10 (diacetylated), PNAG26 (triacetylated) and PNAG30 (tetracetylated) were the strongest binders, along with PNAG31 (fully N-acetylated, however this compound had been found poorly immunogenic in previous reports). However the sentence at lines 255-256 («glycan microarray showed that F598 actually preferred highly acetylated PNAG with the PNAG30 and PNAG31...») is misleading, since, according to Fig. 3A, PNAG26 shows higher Ab binding than PNAG31.

Response: We thank the reviewer for pointing this out. We have revised this part of the text (lines 263-267) as the following:

Interestingly, although mAb F598 was initially identified due to binding to de-acetylated PNAG with only ~ 15% N-acetylation³³, it had little binding to glycan PNAG0 or any glycans containing only one Ac moiety. Highly acetylated PNAG such as PNAG30 and PNAG31 with four or more consecutive GlcNAcs are among the strongest binders (**Figure 3**).

3.. **Comment 3:** The immunization protocol was repeated on rabbits, confirming the results obtained on mice. In a further experiment, the authors evaluated the ability of sera from rabbits immunized with Qbeta-conjugates to recognize native PNAG extracted from bacteria. All sera showed strong binding (the PNAG26 was the best), much higher than the binding observed with the sera from TT-conjugate, demonstrating again the superior performance of the Qbeta carrier. For clarity reason, part C and part D of Fig. 2 should be included in a different figure and moved ahead, closer to the text that describes them. The functional activity of the post-immune sera was evaluated by antibody-mediated complement deposition and opsonophagocytic killing assay.

Response: Following reviewer's suggestion, we have removed Figures 2C and 2D, and incorporated them as Figures 4A and 4B. They are now first mentioned in the text closer to where the figures appear.

Figure 4. Immunization of rabbits with mQβ-PNAG conjugate induced significant anti-PNAG IgG antibodies. A) IgG antibody titers to the immunizing PNAG oligosaccharide in rabbit sera on

day 35 after prime vaccination. B) IgG antibody titers in pooled rabbit sera from mQ β -conjugate or 5GlcNH₂-TT conjugate immunized animals (n=2 per group) as well as titer of natural human IgG in pooled human serum against native PNAG polysaccharide purified from *Acinetobacter baumannii*. The numbers above symbols are the average titer numbers. C) Stacked bar graphs depicting the IgG signals at the serum dilution of 1:50,000 for each rabbit (n = 2) immunized with mQ β -PNAG0, mQ β -PNAG10 and mQ β -PNAG26 as well as pre-immune sera respectively on the array. The complete microarray results are shown in **Data S1**; D) Normalized binding of the comprehensive library of PNAG pentasaccharides by IgG antibodies from post-immune sera of rabbits immunized with mQ β -PNAG0, mQ β -PNAG10 and mQ β -PNAG26, respectively, as well as pre-immune sera. PNAG sequences are grouped together according to the total number of acetamides in the molecules. The color scale bar is shown on the right with 100% indicating the strongest binding to a PNAG component and 0% indicating the weakest binder. For each antigen, the two rows represent sera from two rabbits per group immunized with the specific construct.

4. **Comment 4:** Finally, both active and passive immunization experiments showed that the Qbeta-conjugates are able to provide effective protection of mice against lethal challenges by *S. aureus*. Particularly intriguing is the result described at lines 438-446: a 1:1 mixture of PNAG26 (triacylated) and PNAG0 (non acetylated) sera provided 100% protection to mice against lethal challenges. This result seems to go against the hypothesis of the occurrence of a specific glycoepitope in the synthetic PNAG pentasaccharides. The authors should comment about this and try to provide a plausible interpretation of this result. For the same reason, I also suggest the authors to temper their claim of the introduction (lines 85-87), since no specific PNAG epitopes could be identified in this work.

Response: The general preference for PNAG structure of the mAb F598 has been identified through the microarray study as it prefers PNAG with 2 or more Ac moieties. This has provided very valuable guidance to our vaccine design. We have revised lines 85-88 as the following:

The fine patterns of free amine/NHAc of PNAG oligosaccharides are found to be critical for mAb binding. The structural patterns identified through the microarray study guided the selection of PNAG epitopes for the design for next generation PNAG based vaccine, which can provide highly effective protection in multiple mouse models against *S. aureus* infections, including those by MRSA.

A potential explanation of the higher protective efficacy observed with the combined sera is now provided in the main text (lines 459 - 464) as the following:

The higher protective efficacy observed with the combined sera was presumably because the PNAG polysaccharide can be heterogenous in the amine/acetylation patterns. While some of the sequences such as the fully deacetylated PNAG0 may be rare within the native PNAG polysaccharide, antibodies generated by mQ β -PNAG0 can complement those by mQ β -PNAG26. Thus, the combination of two mQ β -PNAG constructs can broaden bacterial recognition and enhance protection against bacterial challenges.

5. **Comment 5:** As a general comment, in my opinion the manuscript is nicely written, well-organised, clear and easily understandable even for readers non expert in the field. The text is fluent and not verbose. The list of references is exhaustive, in each section the most relevant previous works are appropriately cited.

Response: We thanked the reviewer for the positive feedback regarding our work!

6 **Comment 6:** There is no doubt that the library of synthetic PNAG pentasaccharides is a very valuable tool to guide the identification of protective glycoepitope(s), a key issue in the glycoconjugate vaccines development. However, the disclosed results do not show a clearly defined glycoepitope. In addition, it is unclear whether the superior immunogenicity of the novel neoglycoconjugates is due to the carrier Qbeta only, or (at least in part) to the precise acetylation pattern of the synthetic oligosaccharides. There are no convincing data that allow to dissect the role of the two components. It is well-known that the carrier can have a crucial role in determining the immunogenicity of the glycan payload, for example by exposing the carbohydrate antigens in a better way to bind B cell receptors. The authors should take in account and discuss these aspects both in the introduction and, in particular, in the conclusion. Another issue concerns the carbohydrate loading in the conjugates. According to Fig. S1, the carbohydrate loading in all Qbeta-PNAG conjugates is the same (250 glycans per Qbeta particle). However, in the head-to-head comparison (lines 199-202) equal amounts of Qbeta-PNAG0 and TTHc-PNAG0 conjugates were administered: this means that the two conjugates have the same carbohydrate density? Can the authors comment about this point? More in general, can the authors clarify the role of the carbohydrate loading in their immunogenicity studies?

Response: We think the superior performance of the mQ β -PNAG conjugate can be attributed to two factors: 1) the mQ β carrier as head-to-head comparison has shown that the mQ β -PNAG0 conjugate induced stronger antibody responses compared to the corresponding TTHc-PNAG0 conjugate; 2) the identification of the desired acetylation pattern of the PNAG antigen. This discussion has been added to the text as the following (lines 513 – 521):

In order to enhance the potential protective efficacy, several areas of the PNAG based vaccine can be improved. As carbohydrates are typically T cell independent B cell antigens, an immunogenic carrier is critical. We have demonstrated that mQ β is a powerful carrier. The mQ β -PNAG conjugate was found to be superior in inducing higher levels of anti-PNAG IgG antibodies as compared to the corresponding PNAG conjugate with the TTHc carrier.

Besides the carrier, another important factor in vaccine design is the identification of the protective epitope of the PNAG antigen, which was hampered by the lack of diverse structurally well-defined PNAG compounds.

The reviewer is right that the carbohydrate loading on the carrier can significantly impact the immune responses. The overall densities of the antigen on mQ β -PNAG0 and TTHc-PNAG0 conjugates were similar, and the total amount of antigen used for immunization in the mQ β -PNAG0 and TTHc-PNAG0 were the same. This information has been provided as the following (lines 203 - 217):

To compare with our mQ β -PNAG conjugate, we covalently linked PNAG0 (5GlcNH₂) with the TT heavy chain (TTHc) using SBAP with an average loading of 4.7 PNAG0 per protein molecule (**Scheme 3B** and **Supplementary Figure 2**). The recombinant TTHc is a suitable surrogate of TT³³. As the molecular weight of mQ β particle (2,540 kDa for the protein shell) is about 49 times that of the TTHc (MW ~ 52 kDa), the overall densities of PNAG0 on mQ β -PNAG0 and TTHc-PNAG0 were similar.

Head-to-head comparison between the mQ β and the TTHc-PNAG0 conjugates was carried out. Groups of female C57Bl6 mice (n =5 per group) were immunized with freshly prepared mQ β -PNAG0 (8 nmol corresponding to 8 μ g of PNAG0 per injection) or the TTHc-PNAG0 conjugate (8 nmol PNAG0 per injection) on days 0, 14 and 28. Monophosphoryl lipid A (MPLA, 20 μ g) was added to each vaccination as the adjuvant. A control group of mice received a mixture of mQ β with PNAG0 at equivalent total amounts of mQ β , PNAG0, and MPLA following the same immunization protocol. On day 35, sera were collected from all mice.

Furthermore, as suggested by the reviewer, we have expanded the discussion on the role of carbohydrate loading on immunogenicity as the following (lines 190 - 197):

It is known that the antigen loading density on Q β can significantly impact the levels of antibodies induced against the target antigen^{29,30}. When the loading level of antigen was low (< 50 copies per particle), despite the same total amount of antigen administered, the antibody responses induced were low^{29,30}. Increasing the local density of the antigen on the particle (over 100 antigens per particle) can significantly improve the antibody responses, which is presumably due to the more effective cross-linking of the B cell receptors on B cells³¹. The loading density of PNAG on the mQ β -PNAG conjugate was higher than the threshold antigen level needed for powerful B cell activation.

7) **Comment 7:** In conclusion, in my opinion this is a beautiful piece of work which deserves publication on Nature Communications with the revisions mentioned above.

Additional minor remarks are the following: - line 134: it should be "the glycosylation of acceptor 8 with disaccharide 9..." - Scheme 3, part B. cysteine instead of cycteine

-Figure 4B: correct with mQbeta(symbol)-PNAG (Y-axis) In addition, a) and b) in the caption should be replaced with A) and B)

- lines 406, 407, 413: I think it should be TTHc conjugates

-line 476: which mQbeta conjugate was used in this experiment?

Response: We thank the reviewer for the positive feedback toward our work and for the careful reading of our manuscript. The issues identified have been corrected as the following:

Line 135: the glycosylation of acceptor 8 with disaccharide 9

The typo in Scheme 3B has been corrected.

Figure 4B has been updated as Figure 4D.

TT in lines 419, 420 and 426 have been revised to TTHc.

Line 494, it is mQ β -PNAG26

Reviewer 2

1. **Comment 1:** Overall, the manuscript is clear, and the results of this study will be of broad interest to the audience of Nature Communications. However, the authors should address the following concerns before the manuscript is considered for final publication:

The results presented in Figure 2 for immunization studies are discussed in two separate parts of the manuscript (on page 10 and then on page 14). To avoid discussing these same findings in two separate parts of the manuscript, this reviewer recommends that the authors move the discussion

profiling the PNAG specificity of F598 to appear before the discussion of immunization trials reported currently in Figure 2.

Response: We thank the reviewer for the high opinion of our work. For revision of Figure 2, please see comment 3 of reviewer 1. We have separated Figures 2C and 2D from Figure 2, and moved them to Figure 4 to be closer to where they are discussed in the text.

2. **Comment 2:** Figure 2D. Should be plotted either showing the individual data points or as a box and whisker plot and should indicate the number of samples represented in each data point. The figure legend for figure 2D should also indicate the bacterial source of the native PNAG used in this study.

Response: Figure 2D has become Figure 4B in the revised manuscript. The bacterial source of the native PNAG used has been added to both the figure caption for Figure 4B and the main text.

Figure 4. B) IgG antibody titers in pooled rabbit sera from mQ β -conjugate or 5GlcNH₂-TT conjugate immunized animals (n=2 per group) as well as titer of natural human IgG in pooled human serum against native PNAG polysaccharide purified from *Acinetobacter baumannii*. The numbers above symbols are the average titer numbers.

3. **Comment 3:** Figure S8 should appear before figures S6 or S7 in the supplemental information as it is mentioned in the text first. Also, the authors should include the statistical method used to determine significance in these comparisons.

Response: We thank the reviewer for pointing this out. We have moved supplementary figure 8 before supplementary figures 6 and 7 in the revised SI. The statistical analysis was performed using one way ANOVA. This information is now provided in the caption of this new Supplementary figure 6.

Supplementary Fig. 6. Post-immune sera of A) mice and B) rabbits immunized with the mQ β -PNAG conjugates recognized the *S. aureus* cells well as determined by ELISA. The EC50 values

(the fold of serum dilution that gives half-maximal binding) of the IgG titers were plotted with each symbol representing one animal and the horizontal line is the geometrical mean value of the titers within the group. The ELISA titers were determined against *S. aureus* cell coated ELISA wells. Statistical analyses were performed using one way ANOVA. * $p < 0.05$; ** $p < 0.01$; *** $p < 0.001$; **** $p < 0.0001$.

4. Comment 4: The details for the method used to determine complement deposition (results are reported in Figure 5A) are not described in the manuscript. This method should be described in the materials and methods section in detail. The authors should also clearly indicate what statistical test is being used to determine statistical significance for the data reported in Figure 5A as it is not clear what comparisons the reported P-values refer to. The authors should report individual datapoints for the data shown in this graph.

Response: The details for complement deposition has been added to the experimental section as the following from line 730:

ELISA assays to determine binding to PNAG polysaccharide and analysis of deposition of complement component C1q.

ELISA assays to determine antibody titers to PNAG polysaccharide and analysis of the deposition of complement component C1q were performed as described⁴⁶ except that mouse or rabbit antisera were used in place of equine serum and secondary antibodies used for PNAG titer determinations were specific to either mouse or rabbit IgG. Native PNAG polysaccharide was purified from *A. baumannii* as described^{37,38}. Immulon 4 HBX 384 well plates (ThermoFisher 8755) were coated with purified PNAG polysaccharide. For PNAG coating, plates were treated with 100 μ L of PNAG (0.6 μ g/mL) and then incubated for a minimum of 3 hours. The plates were treated with 100 μ L of methanol (-20° C) for 5 minutes before blocking for 60 minutes at 37° C with 120 μ L of PBS + 1% skim milk (1 g/100 mL) that was heated to 65° C for 60 minutes to pasteurize the solution. All C1 deposition assays were performed in quadruplicate for each test serum. Samples of each serum were heat inactivated (56° C for 30 minutes), and 50 μ L of each serum was combined with 50 μ L of complement source (100 μ L/well) for 60 minutes on a rocker at 37° C. The purpose of heat treating the serum was to inactivate complement in each serum such that, by using a common source of complement, innate differences among sera to deposit complement could not be attributed to innate differences in complement activity in the sera. After washing 3 times with PBS + 0.05% Tween, affinity-purified goat anti-complement C1q primary antibody (Cedarlane, Inc, Burlington, North Carolina) was placed in each well (100 μ L/ well) at a dilution of 1:1000, and incubated for 60 minutes at room temperature. This was followed by 3 wash cycles with PBS + 0.05% Tween, after which an anti-goat IgG-alkaline phosphatase (AP) secondary antibody was placed in each well at a dilution of 1:2000. Plates were incubated for 60 minutes at room temperature. After another 3 wash cycles, the AP-enzymatic production of color was elicited using a 1 mg/mL of 4-nitrophenyl phosphate disodium salt hexahydrate in substrate buffer for 30 minutes at 37° C. The ODs for each plate were recorded at a wavelength of 450 nm. Titers were calculated using simple linear regression wherein the dilution curve crossed the X axis at an OD value equal to approximately 3X the standard deviation of the mean upper OD

limits in control, negative sera. Antisera to the conjugate of 5GlcNH₂ to TT (5GlcNH₂-TT) were prepared in rabbits as previously described^{13,47}.

For Figure 5A, the rabbit sera in this figure were pooled sera from 2 immunized rabbits per group. The normal human serum is a pool of normal sera from hundreds of individuals. The mean and 95% C.I. were from the calculation for the endpoint titer. P values indicate the significance of the deviation of the slope of the titration curve from zero to identify sera with activity at P<0.05. The revised caption for figure 5A is shown below:

Figure 5. Deposition of C1q onto purified PNAG polysaccharide were determined as described⁴⁶. Tests were performed using pooled sera from rabbits (n=2 per group) immunized with mQβ-PNAG conjugates, the 5GlcNH₂-TT conjugate or from a sample of pooled normal human sera. The mean and 95% C.I. were from the calculation for the endpoint titer. P values indicate the significance of the deviation of the slope of the titration curve from zero to identify sera with activity at P<0.05. mQβ-PNAG10 and mQβ-PNAG26 conjugates were more potent than the mQβ-PNAG0 and 5GlcNH₂-TT conjugate in inducing C1q deposition onto purified PNAG. Normal human serum had no C1q depositing activity in spite of having a binding titer to PNAG (see **Figure 4B**) consistent with prior reports that naturally-acquired human antibody to PNAG is not functional due to the inability to activate the complement pathway^{12,34}. Titers were determined by simple linear regression.

5. Comment 5: For the opsonic killing assay results presented in Figure 5B and Figure S6, there appear to be three data points reported for each serum dilution, but according to the figure legend sera from n=2 rabbits were used. The methods also report that duplicate measurements for each serum dilution were measured. Thus, it is not clear what the data points reported in this figure represent and why there are three data points at each serum dilution. The authors should clearly indicate what each data point represents (ie. are these the mean % killing values from duplicate wells, etc...).

Response: We thank the reviewer for pointing this out. We used pooled rabbit sera. The three data points at each serum dilution were from three aliquots of the sera. This info has been added to the caption for figure 5B and the experimental section. The revised caption for figure 5B is shown below:

B) Pooled sera from rabbits (n =2 per group) immunized with mQβ-PNAG conjugate led to significantly higher levels of opsonic killing activities against *S. aureus* cells. Three aliquots were prepared from each pooled serum and the individual values of the three aliquots were presented.

6. Comment 6: Figure 7A/B. The authors need to indicate the number of mice used in each treatment group. This is indicated in Figure 7C/D but not in A/B.

In Figure 7D the t-test is not the appropriate statistical test to evaluate this data. Instead, a one-way ANOVA should be used, or since the distribution of the data appears to be bimodal (ie. there appears to be two distinct populations in each data set) a more appropriate analysis may be a Kruskal-Wallis non-parametric test.

Response: There were 10 mice per group for Figures 7A and 7B. This info has been added to the caption. One-way ANOVA and the Kruskal-Wallis non-parametric test have been performed on the data. This info has been added to the figure caption and the experimental section. The revised caption for Figure 7 is shown below:

Figure 7. Transfer of antisera from mQβ-PNAG immunized rabbits to mice A) provided

significant protection to mice (n = 10 per group) against the lethal challenges by *S. aureus* ATCC 29213. Statistical analysis was performed with the logrank test. * $p < 0.05$; *** $p < 0.001$; **** $p < 0.0001$; and B) significantly reduced bacterial count in mouse kidney. The combination of sera from mQ β -PNAG0 and mQ β -PNAG26 immunized rabbits provided complete protection to mice. Statistical analysis for survival was performed using the logrank test. Analysis of *S. aureus* cfu/gm was by Kruskal-Wallis nonparametric ANOVA ($P < 0.0001$ for overall effect of serum given). P values for pairwise comparisons shown on graph by Dunn's multiple comparisons test. Transfer of antisera from mQ β -PNAG immunized rabbits to mice C) provided significant protection to mice against the lethal challenges by MRSA strain 1058 (n = 10 per group); and D) significantly reduced bacterial count in mouse kidneys. Sera from mQ β -PNAG26 immunized rabbits provided the highest protection to mice. The horizontal line represents the median value of each group. Statistical analysis for survival was performed using the logrank test. Analysis of MRSA cfu/gm was by Kruskal-Wallis non-parametric ANOVA ($P = 0.0967$). P values for pairwise comparisons shown on graph by Dunn's multiple comparisons test.

7. Comment 7: A table detailing the bacterial strains used in this study and their sources should be include in the supplemental information of this article.

Response: A table detailing the bacterial strains has been added to the supplemental information as Supplementary Table 1.

Supplementary Table 1. Staphylococcus strains used in this study.

strain	description	reference
41	JE2; Laboratory derived wild type parental MRSA; USA300_LAC; CC8	2
954	icaA::Tn; NTML NE37 bursa aurealis transposon (Tn) mutant, Erm ^R	2
clinical isolates		
1055	MRSA abscess hand cellulitis	3
1056	MRSA abscess left arm	3
1057	MRSA left wrist/ index finger	3
1058	MSSA abscess left foot osteomyelitis	this study
1059	MSSA bone from the coccyx, chronic osteomyelitis	3
1153	MRSA hallux bone	this study

8. Comment 8: Figure S5. The authors should provide a more descriptive figure caption to clearly indicate the results observed in these studies. Bright field or phase contrast images for the fields of view presented in Figure S5 should also be provided to demonstrate that the samples contain similar amounts of bacterial cells.

Response: We have added more descriptive details to figure caption. Bright field images have been added to Supplementary Figure 5b. Unfortunately, we did not have the bright field images for Supplementary Figure 5a. All samples contained similar numbers of bacterial cells. The revised Supplementary Figure 5 is shown below:

Supplementary Fig. 5. A) *S. aureus* ATCC 29213 cells were stained by pre-immune, and post-immune sera from rabbits immunized with mQβ-PNAG0, mQβ-PNAG10, and mQβ-PNAG26 respectively followed by the addition of Alexa Fluor 594 labeled goat anti rabbit IgG Fc 2nd antibody. The binding was visualized through fluorescence microscopy. Sera from rabbits immunized with mQβ-PNAG10, and mQβ-PNAG26 showed significant binding to the bacteria. The scale bar is 10 μm. B) Fluorescence images and bright field images of MRSA strains, i.e., 41, 1055, 1056, 1057, 1058, 1059, and 1153 upon staining with immune sera from rabbits immunized with mQβ-PNAG0, mQβ-PNAG10, and mQβ-PNAG26 respectively. The post-immune sera bound with these strains well as detected by the Alexa Fluor 594 labeled goat anti-rabbit IgG Fc 2nd antibody. In comparison, a control strain lacking PNAG expression with *icaA* gene knock out

(954) showed negligible binding by the sera, suggesting the recognition is PNAG dependent. Scale bar: 10 μm .

9. Comment 9: Figure S7. The authors should fully write out all of the abbreviations used in the figure caption.

¹³C NMR data are reported for PNAG0-PNAG32 but the spectra are not provided as part of the supplemental information. The authors should provide these spectra in the SI if available.

Response: Abbreviations have been defined in the caption of Supplementary Figure 8 (Figure S7 in the original manuscript). ¹³NMR data were obtained from HSQC spectra, which are provided in the SI. The revised caption for Supplementary Figure 8 is shown below:

Supplementary Fig. 8. Non-Metric Multidimensional Scaling (NMDS) analysis of bacterial communities of controls and treatments showing no significant changes in the gut microbial community following mQ β -PNAG26 immunization. T1 and T2 refer to the two sampling times. **T1 = first sampling right before immunization, T2 = control and treatment samples at 42 days after the prime immunization.** Each of the four groups is delineated by a convex hull describing a minimum area. The stress test values for 2D and 3D NMDS were 0.141 and 0.109, respectively. Amplicon sequence variants (ASV) datasets revealed similar patterns of ordination. When comparing treatment versus control within a timepoint, ANalysis Of SIMilarity (ANOSIM) and Permutational multivariate analysis of variance (PERMANOVA) values of -0.04 (T1) and 1.518 (T2) indicated high similarity in the phylogenetic structure of the bacterial communities.

REVIEWERS' COMMENTS

Reviewer #1 (Remarks to the Author):

This is the second revision of this manuscript and, after reading the authors' responses to my comments, I confirm that my concerns have been fully addressed in the revisions, and all clarifications required have been convincingly provided.

Therefore, I recommend the publication of the revised manuscript in Nature Communications

Reviewer #2 (Remarks to the Author):

The authors have addressed all of my comments adequately.